# Understanding the Causes of Satellite–Model Discrepancies in
# Aerosol–Cloud Interactions Using Near-LES Simulations of Marine
# Boundary Layer Clouds
Shaoyue Qiu[1], Xue Zheng[1], Peng Wu[2], Hsiang-He Lee[1], and Xiaoli Zhou[3]
[1]Atmospheric, Earth, and Energy Division, Lawrence Livermore National Laboratory, Livermore, CA, U.S.A.
[2]Atmospheric, Climate, & Earth Sciences Division, Pacific Northwest National Laboratory, Richland, WA 99354,
Washington, USA.
[3]Department of Physics and Atmospheric Science, Dalhousie University, Halifax, NS, Canada
*Correspondence to*: Xue Zheng (zheng7@llnl.gov)
Submitted to Atmospheric Chemistry and Physics   17 July 2025
**Abstract.** Aerosol–cloud interactions (ACI) remain the largest source of uncertainty in model
estimates of anthropogenic radiative forcing, primarily because of deficiencies in representing
aerosol–cloud microphysical processes that lead to inconsistent cloud liquid water path (LWP)
responses to aerosol perturbations between observations and models. To investigate this
discrepancy, we conducted a series of large-eddy scale simulations driven by realistic
meteorology over the Eastern North Atlantic, and evaluated LWP susceptibility, precipitation
processes, and boundary layer thermodynamics using satellite and ground-based observations.
Simulated LWP responses show a strong dependence on cloud state. Non-precipitating thin
clouds exhibit a modest LWP decrease with increasing cloud droplet number concentration ($N_d$),
consistent in sign but weaker in magnitude than satellite estimates, reflecting enhanced turbulent
mixing and evaporation. The largest model-observation discrepancy occurs in non-precipitating
thick clouds, where simulated LWP susceptibilities are strongly positive ($+0.32$) while
observations indicate large negative values ($-0.69$). This discrepancy stems from excessive
precipitation driven by underestimated entrainment, overly active accretion, and overly broad
drop-size distributions in polluted conditions. While our high-resolution setup mitigates the
excessive drizzling common in coarser models and captures key regime transitions, these biases
persist—highlighting that improved parameterization of cloud-top processes, precipitation, and
aerosol effects are needed beyond simply increasing model resolution.
Additionally, misrepresented moisture inversions in reanalysis introduce a moist bias in cloud-
top relative humidity, further amplifying positive LWP susceptibility. Our results also suggest
that large negative $N_d$–LWP relationships in observations may reflect internal cloud processes
rather than true ACI effects.

## 1. Introduction

Marine boundary layer clouds exhibit substantial influence on Earth's radiation balance due to their high albedo and extensive global coverage. Aerosols modulate cloud albedo through changing cloud droplet number concentration ($N_d$), cloud liquid water path (LWP), and cloud fraction. The estimated radiative cooling from aerosols partially offset the warming from greenhouse gas emission (Slingo 1990). However, aerosol-cloud interaction (ACI) remains the most uncertain component of anthropogenic radiative forcing (Foster et al., 2021). In particular, liquid-phase cloud adjustments in LWP, cloud fraction, and cloud lifetime present the largest uncertainties in determining the net radiative forcing of ACIs, especially under varying large-scale conditions (Han et al., 2002; Small et al., 2009).

Among these uncertainties, the LWP response to aerosol perturbations has drawn particular attention due to its large spread in both observations and numerical model simulations. Theoretically, increasing aerosols would reduce droplet size and suppress precipitation, thereby increasing LWP and cloud lifetime (Albrecht, 1989). However, smaller droplets might also enhance evaporation and entrainment, leading to a reduced LWP in non-precipitating clouds (Ackerman et al., 2004; Xue and Feingold, 2006; Bretherton et al., 2007). This competition between processes leads to a bifurcated LWP response that varies with aerosol concentration, cloud type, and background meteorology.

In recent years, numerous satellite studies have reported an overall decrease of LWP with increasing $N_d$ for non-precipitating clouds in polluted environments and an increase in LWP for precipitating clouds (e.g., Gryspeerdt et al. 2019, 2021; Toll et al., 2019; Zhang et al., 2022, 2023; Qiu et al., 2024; Yuan et al., 2023; 2025). In contrast, current global climate models (GCM) mostly simulate a positive LWP response to aerosol perturbation regardless the cloud conditions, which leads to an over-estimation of the aerosol-induced radiative forcing that is dominated by ACI (e.g., Ghan et al., 2016; Michibata et al., 2016; Mülmenstädt et al., 2024). This discrepancy could stem from the poorly resolved cloud processes in GCM due to its coarse horizontal resolution (~100 km). Recent development in computing have enable the global convection-permitting models (GCPMs) with kilometer-scale grid spacing, serving as an invaluable complement to the traditional climate models (e.g. Satoh et al., 2019; Stevens et al., 2019; Caldwell et al., 2021; Donahue et al., 2024). Notably, Sato et al. (2018) employed a GCPM and simulated a negative LWP response, attributing it primarily to better resolved evaporation and condensation processes from aerosol perturbations. Yet, other CPM studies with finer resolution than Sato et al. (2018) mostly simulate an increase in LWP with aerosol perturbations (e.g., Fons et al., 2024; Christensen et al, 2024), largely due to uncertainties in microphysics schemes, particularly regarding the treatment of precipitation (White et al., 2017).

Since most current GCPMs and GCMs adopt two-moment microphysics schemes, it is important to evaluate the precipitation parameterization in these schemes with observational constraints, in addition to the influence of precipitation process on the simulated ACI. Meanwhile, Terai et al. (2020) found that the lack of decrease in LWP in kilometer-scale models could be due to the lack of resolving the sub-kilometer processes that are most relevant to ACI processes. For example, they found that when increasing model resolution from 4 km to 250m, the fraction of precipitating clouds largely decreases, especially for thick clouds, and the LWP response becomes negative for non-precipitating clouds. Therefore, it is critical to assess the benefit of increasing model resolution to near large-eddy simulation (LES) scale in representing precipitation, as well as the evaporation-entrainment feedback responsible for LWP reduction

without altering the structure of the microphysics parameterization and ultimately reconcile the
LWP adjustment observed by satellite with those estimated by GCM and GCPM.
With model resolutions ranging from 25 m to 200 m, numerous LES studies have utilized
idealized meteorological conditions and have provided valuable process-level understanding on
the mechanisms governing cloud responses to aerosol perturbations (e.g., Xue and Feingold,
2006; Xue et al., 2008; Bretherton 2007; Seifert et al., 2015; Glassimire et al., 2019; Hoffman et
al., 2020; Chen et al., 2024; Zhang et al., 2024). However, ACI and cloud processes using
idealized simulations cannot be directly evaluated or constrained by observations, limiting their
ability to explain the divergent LWP response between the two. Additionally, many LES studies
are conducted with limited domain size, which cannot resolve mesoscale organization and
variability of cloud and precipitation, both of which have been shown to significantly affect
retrieved $N_d$-LWP relationships (e.g., Zhou and Feingold; 2023; Kokkola et al., 2025; Tian et al.,
2025). Finally, both aerosol and cloud fields are strongly modulated by synoptic conditions (e.g.,
Engström and Ekman, 2010, Zheng et al. 2011, Zheng et al. 2025). LES studies focused on a
small number of cases fail to capture the influence of cloud regimes and synoptic variabilities on
ACI, both of which determine the magnitude and sign of cloud responses to aerosol
perturbations.
The Eastern North Atlantic (ENA) region is uniquely suited to address this issue due to
its location at the transition between midlatitude and subtropical regimes, experiencing various
synoptic conditions and cloud regimes (e.g., Remillard & Tselioudis, 2015; Zheng et al., 2025).
In addition, the ENA region and the availability of long-term, high-quality ground-based
observations from the DOE Atmospheric Radiation Measurement (ARM) program make it
possible for process-level evaluation with the comprehensive observations. Marine boundary
layer (MBL) clouds in this region are frequently drizzling and sensitive to aerosol and
meteorological perturbations, making them ideal for studying aerosol-cloud-precipitation
interactions (Wood et al., 2015).
*The goal of this study is to evaluate key ACI processes, such as precipitation suppression*
*and evaporation-entrainment feedback, as well as precipitation treatment in a two-moment*
*scheme, through simulations approaching LES scales.* To address limitations in previous LES
studies, we perform a series of simulations using a nested-domain configuration to seamlessly
simulate the realistic circulations across different synoptic regimes, with the innermost domain
spanning 1°×1°, consistent with typical GCM grid spacing and the spatial scale used in satellite
observations to quantify $N_d$-LWP relationships (introduced in Sect. 2). To investigate the
variation of ACI across different synoptic conditions, we simulated an ensemble of realistic
MBL cloud cases across three synoptic regimes, each characterized by northerly surface flow
over the ENA site. The classification of synoptic regimes is based on our previous study (Zheng
et al., 2025), in which seven major synoptic regimes were identified using both surface and mid-
level meteorological data. To enable a process-level evaluation of the parameterization of the
warm rain process, we leverage ground-based radar measurement from the DOE ARM ENA site
and apply a newly developed radar simulator for direct model-observation comparison.

## 2. Data and Methodology

### 2.1 Datasets

This study adopts both satellite and ground-based observations to assess the simulated cloud, precipitation processes, and ACI processes. For satellite observations, we used cloud retrievals derived from the Spinning Enhanced Visible InfraRed Imager (SEVIRI) on the geostationary satellite Meteosat-10 and Meteosat-11 over the ENA region. The cloud retrievals are based on the methods developed by the Clouds and the Earth's Radiant Energy System (CERES) project using the Satellite ClOud and Radiation Property retrieval System (SatCORPS) algorithms (Minnis et al., 2011, 2021; Painemal et al., 2021). The SEVIRI Meteosat cloud retrieval products are pixel-level cloud retrievals produced by NASA LaRC SatCORPS group, specifically tailored to support the ARM program over the ARM ground-based observation sites. For Meteosat-10 and Meteosat-11 cloud retrievals, they have a spatial resolution of 4-km and 3-km at nadir and an hourly and half-hourly temporal resolution, respectively.

In this study, we used the cloud mask, cloud effective radius ($r_e$), cloud optical depth ($\tau$), LWP, cloud phase, and cloud top height variables in the SEVIRI Meteosat cloud retrieval product (Minnis et al., 2011, 2021). We focus on warm boundary layer clouds with cloud top below 3km and a liquid cloud phase. The $r_e$ and $\tau$ retrievals are based on the shortwave-infrared split window technique during the daytime. Cloud LWP is derived from $r_e$ and $\tau$ using the equation: $LWP = \frac{4 r_e \tau}{3 Q_{ext}}$, where $Q_{ext}$ represents the extinction efficiency and assumed constant of 2.0. Cloud mask algorithm is consistent with the CERES Ed-4 algorithm, as described in Trepte et al. (2019), where cloudy and clear pixels are distinguished based on the calculated TOA clear-sky radiance. Cloud top height is derived from the retrieved cloud effective and top temperature, together with the boundary-layer temperature profiles and lapse rate, as described in Sun-Mack et al. (2014). Cloud $N_d$ is retrieved based on the adiabatic assumptions for warm boundary layer clouds, based on the following equation:

$$N_d = \frac{\sqrt{5}}{2\pi k} \left( \frac{f_{ad} c_w \tau}{Q_{ext} \rho_w r_e^5} \right)^{1/2} \tag{1}$$

In Equation (1), $k$ represents the ratio between the volume mean radius and $r_e$, and it is assumed to be constant of 0.8 for stratocumulus, $f_{ad}$ is the adiabatic fraction, $c_w$ is the condensation rate, $Q_{ext}$ is the extinction coefficient, and $\rho_w$ is the density of liquid water (Grosvenor et al., 2018).

To facilitate a consistent comparison, the satellite retrievals are adjusted to the same domain size as the simulation (e.g., 1° × 1°) and the pixel-level cloud retrievals are smoothed to 25-km resolution to reduce impact from cloud heterogeneity and small-scale covariability on the estimated cloud susceptibility (e.g. Arola et al. 2022; Zhou and Feingold, 2023). In the context of ACI: cloud susceptibility quantifies how sensitive a cloud property responds to change in aerosol concentration or $N_d$. To constrain the spatial-temporal variation in meteorological conditions and cloud properties, cloud susceptibility is estimated as the regression slope between $N_d$ and cloud properties within the 1° × 1° domain at each time step of satellite observations. In this study, we quantify LWP and cloud fraction (CF) susceptibilities. Because of the non-linear relations between LWP and $N_d$, the LWP susceptibility is quantified in logarithm scale as $dln(LWP)/dln(N_d)$ (e.g., Gryspeerdt et al. 2019; Qiu et al., 2024), whereas CF susceptibility is quantified as $dCF/dln(N_d)$ (e.g., Kaufman et al. 2005; Chen et al., 2022; Qiu et al., 2024). Due

to the dependence of cloud responses on cloud regimes (e.g., Chen et al., 2014; Zhang et al.,
2022; Qiu et al., 2024), the estimated cloud susceptibilities are displayed in the $N_d$-LWP
parameter space as the classification of cloud states.
In addition to the satellite retrievals, we adopt the ground-based observation at the ARM
ENA site. Specifically, we use the ground-based cloud radar and lidar observations for process-
level evaluation of modeled precipitation processes. In this study, the radar reflectivity ($Z_e$) and
cloud boundaries are from the Active Remote Sensing of Clouds (ARSCL) value added product
(Clothiaux et al., 2001). To remove noise in the data, we smoothed the 4s reflectivity profiles
into 1-minute.  Cloud top height is derived as the upper most range gate height with radar
reflectivity greater than the sensitivity threshold of the Ka-band zenith radar (-40 dBZ) combined
with the hydrometer layer top data in the ARSCL. Cloud base height is from the best-estimate
cloud base height variable in the ARSCL product. Thermodynamic profiles are derived from the
radiosonde data, which is launched at the ENA site twice daily at 0000 UTC and 1200 UTC.
The ground-based $r_e$ and $\tau$ retrievals are based on the parameterization developed in
Dong et al. (1998), where $r_e$ is retrieved from a radiative transfer model as described in Dong et
al. (1997) and parameterized as a function of cloud LWP, shortwave transmission ratio, and
cosine of solar zenith angle. Cloud LWP is retrieved from the brightness temperature measured
by the three-channel microwave radiometer (MWR3C) at 23.8, 30, and 90 GHz (Cadeddu et al.,
2013). The shortwave transmission ratio is calculated from the unshaded pyranometer from the
QCRAD product (Long and Shi, 2006), defined as the ratio between cloudy and clear-sky
shortwave irradiance.
Meteorological and thermodynamic variables are extracted from the European Center for
Medium-Range Weather Forecasts (ECMWF) ERA5 reanalysis data and used as the forcing for
the simulation. ERA5 is the fifth generation of the ECMWF reanalysis, replacing the ERA-
Interim reanalysis. ERA5 provides the best-estimate of the global atmosphere, land surface, and
ocean waves with a horizontal resolution of 31 km and an hourly output throughout (Hersbach et
al., 2020). Atmospheric variables are available on 137 vertical levels, ranging from 1000 hPa
(near surface) to 1 Pa (~80km).

## 2.2 WRF Model

We used the Weather Research and Forecasting (WRF) model version 4.4.2 (Skamarock
et al., 2021) for our simulations. In a companion study, Lee et al. (2025) used the WRF model at
near LES scale with interactive chemistry and aerosol schemes (WRF-Chem) and investigated
ACI and its feedback on both clouds and aerosols in the ENA region. As the WRF-Chem
simulations are 5-10 times more computationally expensive, the present study adopted the same
dynamical and physical configuration and conducted more experiments with prescribed aerosol
concentrations and realistic meteorology.
We employed four one-way nested domains in the model, with the domain size of
27° × 27°, 9° × 9°, 3° × 3°, and 1° × 1°, and spatial resolution of 5km, 1.67 km, 0.56 km, and
190m, respectively, for d01, d02, d03, and d04 domain. The innermost domain (d04) exhibit a
domain size close to most GCM grid spacing and is consistent with the spatial scale for
quantification of cloud susceptibility in satellite study (e.g., Zhang et la., 2022, 2023; Qiu et al.,
2024). The spatial resolution of 190m is much higher than the CPMs and close to the LES scale.
All the analyses and evaluations in this study are based on output from the innermost domain
(d04). There are 75 vertical levels in the model with a model top of ~20 km, the grid spacing is
log-stretched with higher resolution of ~50 m near the surface and increases to ~150 m at the
height of ~1500m . As mentioned above, the initial and lateral boundary conditions for the outer
domain are taken from the ERA5 reanalysis data.
The simulations are performed using the Rapid Radiative Transfer Model for Global
Climate Models (RRTMG; Mlawer et al., 1997), and the Noah land surface model (Chen and
Dudhia 2001). The Mellor–Yamada–Janjic (MYJ; Mellor and Yamada, 1982) planetary
boundary layer (PBL) scheme and the shallow cumulus schemes (Hong and Jiang, 2018) are
utilized for the outer domain (d01 and d02) only. Simulations in this study employ a two-
moment Morrison microphysics scheme, which has been widely implemented in both CPMs and
GCMs (Morrison et al., 2005; Morrison and Gettleman, 2008; Golaz et al., 2022). In the
Morrison two-moment microphysics scheme, the DSD ($\phi$) is defined as:

$$\phi(D) = N_0 D^\mu e^{-\lambda D}, \tag{2}$$

$$\eta = 0.0005714 N_d + 0.2714, \tag{3}$$

$$\mu = \frac{1}{\eta^2} - 1, \tag{4}$$

$$\lambda = [\frac{\pi \rho N_c \Gamma(\mu+4)}{6 q_c \Gamma(\mu+1)}]^{1/3}, \tag{5}$$

where D is the diameter, $N_0$ is the intercept parameter, $\mu$ is the shape parameter, $\lambda$ is the
slope parameter, $\eta$ is the dispersion parameter which governs the width of the DSD (Morrison
and Gettleman, 2008).
Instead of prescribing a constant cloud droplet number concentration, total aerosol
number concentrations are prescribed as a constant throughout the domain with no explicit
vertical variation or transport in all simulations. Aerosol activation follows the parameterization
of Abdul-Razzak & Ghan (2000), with fixed assumptions for size distribution, chemical
composition, aerosol type, and mixing state. The activated fraction mainly depends on the local
supersaturation and updraft speed. The fixed aerosol field neglects spatial and temporal
variability driven by emissions, long-range transport, wet scavenging, and CCN reactivation
from evaporated raindrops. These missing processes can sustain higher CCN concentrations,
suppress precipitation, and potentially exaggerate positive LWP responses.
Despite this simplification, our companion WRF-Chem study (Lee et al., 2025) shows
that, even with full aerosol microphysics, wet scavenging, and aerosol reactivation, the simulated
LWP responses remain broadly consistent with the results presented here, especially the positive
susceptibility in precipitating clouds. This agreement suggests that the key findings of this work
are robust, although the prescribed-aerosol assumption may still contribute to some of the
quantitative discrepancies discussed in Section 3.
For each case, we run the model for 36 hours (except for the consecutive case on 21 July
2016, where the model was run for 60 hours), starting at 12:00 UTC of the previous day and the
first 12 hours are used as model spin-up period. The time resolution of the model is 30 seconds
in the outer domain for advection and physics calculation and is 1 second for the innermost
domain. Model variables are output instantaneously for every 10 minutes for the innermost
domain, similar as in satellite observation of snapshots.
To access the cloud responses to aerosol perturbations, we conduct three sets of
simulations with different prescribed aerosol number concentration of N=100, 500, and 1000
$cm^{-3}$ for all 11 cases. Cloud susceptibility is quantified as the change in domain-mean cloud
properties within the innermost domain at the same output time, comparing polluted and clean
simulations (e.g. N=1000 vs. N=100, N=500 vs. N=100, and N=1000 vs. N=500). With constant
and uniform aerosol concentration, the $N_d$-LWP relations resulting from internal cloud processes
are able to be quantified within each experiment at the same output time. To minimize $N_d$-LWP
relations from cloud heterogeneity and small-scale covariability and to be consistent with the
quantification of cloud susceptibility in satellite observations, the pixel level model outputs are
smoothed to 25-km resolution and $N_d$-LWP relations are quantified as $dln(LWP)/dln(N_d)$
using the smoothed data.
To directly compare the WRF simulations with ground-based observations, we used the
Cloud Resolving Model Radar Simulator (CR-SIM; Oue et al. 2020). It is a forward-modeling
framework which uses consistent microphysics assumptions as in the atmospheric model (i.e.,
the two-moment Morrison scheme in this study) and emulates radar and lidar observables. Some
common radar and lidar variables include: the radar reflectivity factor at horizontal and vertical
polarization, depolarization ratio, Doppler velocity, spectrum width, lidar backscatter, attenuated
backscatter, lidar extinction coefficient, and so on. In this study, we analyzed the simulated radar
reflectivity factor to characterize cloud and precipitation properties.
To distinguish different precipitation modes and the microphysical growth processes that
transition clouds from non-precipitating to drizzling and raining, we investigate the vertical
transition from cloud to precipitation using the Contoured Frequency of Optical Depth Diagram
(CFODD) method (Suzuki et al., 2010) from both observations and model simulations. The
CFODD analysis calculates the frequency of radar reflectivity profiles as a function of in-cloud
optical depth ($\tau_d$), where $\tau_d$ is calculated based on an adiabatic-condensation growth model and
it starts at zero at cloud top and increases downward. One benefit of the CFODD analysis is that
the slope of reflectivity directly relates to the droplet collection efficiency, where the slope of
reflectivity in the common geometric height depends on cloud water content (Suzuki et al.,
2010).

**2.3 Case Studies**

With the focus of MBL clouds in this study, cases are selected when both satellite and
ground-based observations define MBL clouds in the ENA region. For cloud type classification
in ground-based observations, we used the same method as in Zheng et al. (2025), where clouds
are classified into seven types based on the boundaries and duration of each cloud object. In this
study, we include both cumulus and stratocumulus clouds. Days are excluded when only shallow
cumulus clouds are detected to filter out clouds that are below the detectable resolution of the
Meteosat observations and to minimize uncertainties in the cloud microphysical retrievals from
the ground-based observations. We further exclude days with more than three layers of cloud in
the boundary layer to minimize uncertainty in cloud retrievals. Classification of cloud type in
Meteosat observations uses a similar method as the ground-based observations. Cloud objects are
defined as connected cloudy pixels, where low clouds are defined as clouds with 90[th] percentile
of cloud top height below 3km. Low clouds are further classified as stratiform clouds and
cumulus or broken stratiform clouds using an area threshold of 10,000 km$^2$ (Qiu and Williams,
2020).
We focus on summer months (June, July, August) in the ENA region, when this region is
often dominated by the Bermuda high-pressure systems and MBL clouds have the highest
occurrence frequency (e.g., Li et al., 2011; Mechem et al., 2018; Dong et al., 2014, 2023).
Previous studies found that the ARM measurements at the ENA site –located near the northern
shore of the Graciosa Island, the northernmost island in the Azores archipelago – can be
influenced by local emissions and island effects during southerly wind conditions. These impacts
include modification to the aerosol and CCN concentrations, boundary layer turbulence, and the
cloud field (e.g., Ghate et al, 2021, 2023). To minimize these influences, we focus on the three
synoptic regimes identified in Zheng et al. (2025) when the ENA site is influenced by northerly
surface wind: the high-ridge regime (characterized by a mid-tropospheric ridge and surface high-
pressure system), the post-trough regime, and the weak trough regime (Table S1, Figure S1).
With the case selection criteria discussed above, there are a total 11 cases for the WRF
simulations, covering different cloud states and synoptic conditions. The general characteristics
of the 11 cases are listed in Table S1. The synoptic pattern for each case from ERA5 is shown in
Figure S1, the cloud fields observed from Meteosat are shown in Figure S2. WRF simulated
cloud fields in the N=100 and N=1000 experiments are shown in Figure S3, S4. To better
illustrate the large-scale cloud organization and compared with Meteosat observations, the
simulated LWP in domain 2 are shown. As seen in Figures S2-4, our WRF simulations well
capture the frontal systems and synoptic pattern of cloud fields across different cases.
**3 Results:**
**3.1 Case Study: Impacts of Aerosols on PBL Thermodynamics and Cloud Evolution**

Previous studies have demonstrated the distinct cloud responses to aerosol perturbations
between precipitating and non-precipitating regimes in both model simulations and observations
(e.g., Chen et al., 2014; Sato et al., 2018; Gryspeerdt et al., 2019; Fons et al., 2024; Qiu et al.,
2024). To explore these differences, we analyze two representative cases in our simulations: one
dominated by precipitating clouds and another by non-precipitating clouds, to highlight the
distinct interactions among aerosols, clouds, and PBL thermodynamics in the presence and
absence of precipitation.
On 21 July 2016, the ENA site was presented by precipitating stratocumulus clouds from
00:00 UTC to 13:00 UTC, as seen from radar reflectivity profiles in Figure S5b. The clouds
dissipated from 12- 18 UTC and redeveloped after 18 UTC (Figure 1a, black line). The sounding
observations show a moist and well-mixed boundary layer, with relative humidity (RH) near
saturation above cloud top (Figure S6). Our simulation captures the structure of the boundary
layer, with a moist layer above the cloud, and the cloud-top RH close to sounding observations
(99% and 96%, Figure S6c). Due to biases in the ERA5 reanalysis in representing the
temperature inversion, the boundary layer top in the model is ~500m lower than in sounding data
(Figure S6). Consequently, the simulated cloud tops are ~300–500 m lower than both satellite
and ground-based radar observations (Figure 1b, Figure S6).

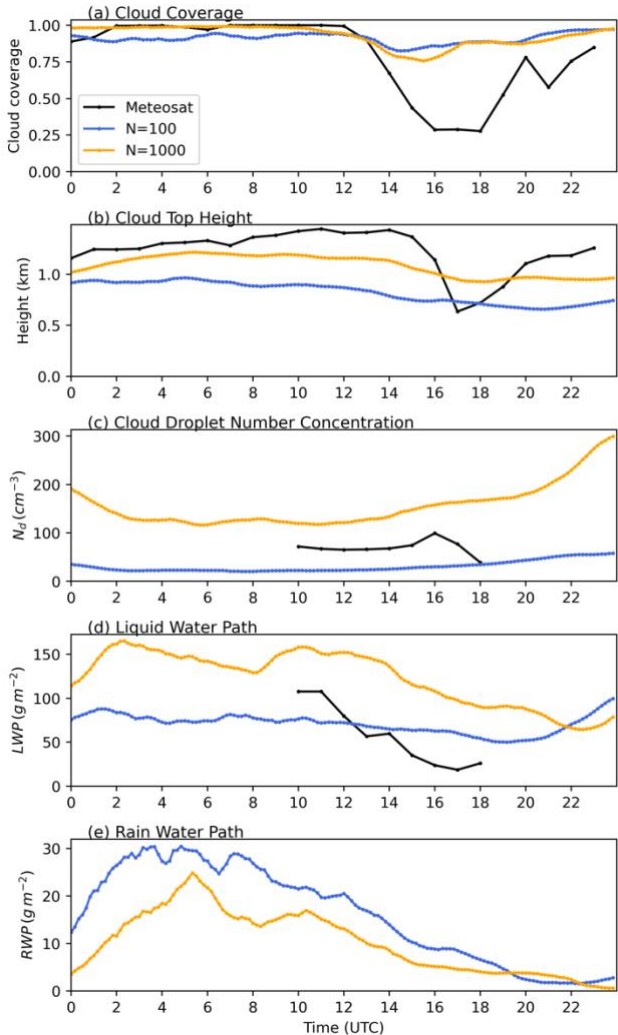


Figure 1. Time series of domain-averaged cloud properties from satellite observations and model
simulation on 21 July 2016. (a) Cloud coverage, (b) cloud top height, (c) cloud liquid water path,
and (d) rain-water path for N=100 (blue lines) and N=1000 (orange lines) experiments.

In the N=100 simulation, WRF model reproduces the overcast and precipitating

stratocumulus clouds, with a domain mean cloud cover varies between 0.90 to 0.94 from 00-13
UTC, which is slightly below that from Meteosat of 0.97 to 1.0 (Figure 1a, blue and black lines).
However, unlike observations, the simulated clouds do not dissipate after 14 UTC; both cloud
cover and LWP remain nearly constant throughout the day (Figures 1a, d, blue lines).  With
increased aerosol concentration (N=1000), the simulated precipitation is suppressed (Figure 1e),
and the cloud layer remains overcast while deepening, accompanied by rising cloud tops and
increasing LWP (Figure 1b, c, orange lines). This cloud response arises from aerosol-induced
precipitation suppression and the corresponding changes in boundary layer processes, as
illustrated in Figure 2. The turbulent kinetic energy (TKE) is calculated as $\frac{1}{2}\left(\overline{u'^2} + \overline{v'^2} + \overline{w'^2}\right)$,
with a unit of $m^2 s^{-2}$, and buoyancy flux is calculated as $g/\theta_0\overline{w'\theta_v'}$, with a unit of $m^2 s^{-3}$.

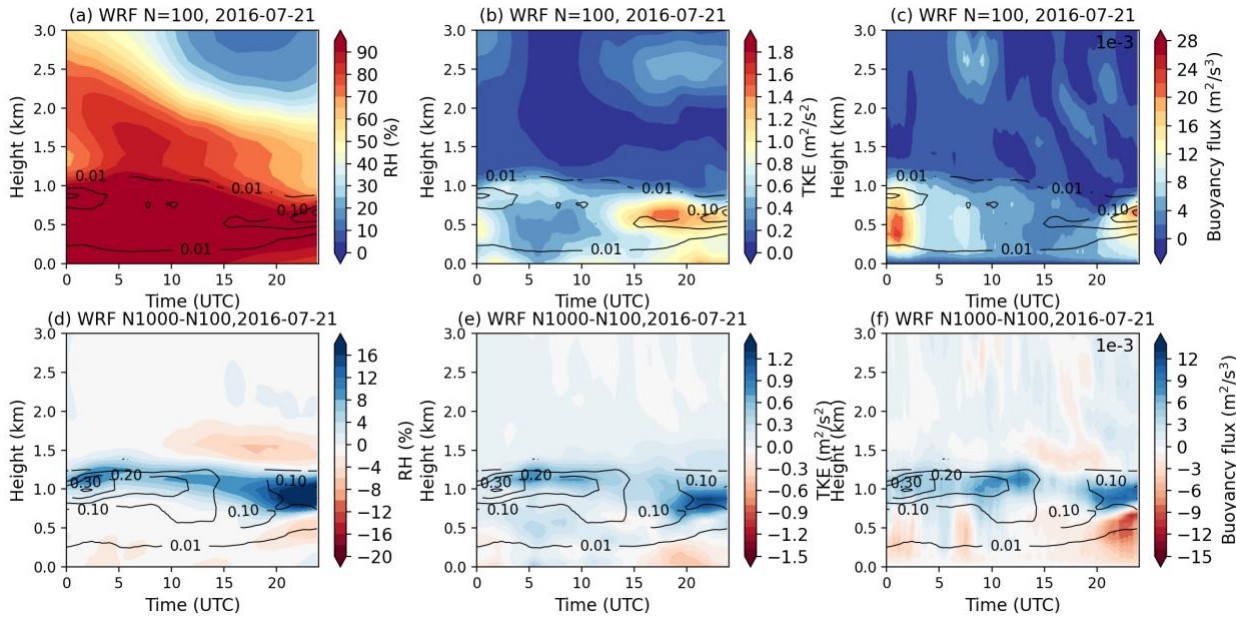

Figure 2. Time series of domain-averaged thermodynamic profiles on 21 July 2016, for (a)
relative humidity, (b) turbulent kinetic energy (TKE) (unit: $m^2 s^{-2}$), (c) buoyancy flux (unit:
$m^2 s^{-3}$) in N=100 simulations, (d) changes in relative humidity profiles, (e) changes in TKE, (f)
changes in buoyancy flux between N=100 and N=1000 simulations. The black contours are
cloud water mixing ratio (unit: g/kg) in (a)-(c) N=100 and (d)-(f) N=1000 simulations.
In the simulations, increases in aerosol concentrations lead to higher $N_d$ and smaller drop
size. As the two-moment Morrison scheme does not consider the cloud drop size in the
parameterization of evaporation, aerosol impacts on clouds and boundary layer occur through the
influence of precipitation on PBL structure. Specifically, aerosols suppress precipitation by
reducing autoconversion with increasing $N_d$, decreasing sedimentation rate and terminal velocity
from smaller droplets. The formation of drizzle release latent heat and reduce both entrainment
and the production of turbulent kinetic energy (TKE) by buoyancy; while the evaporation of
drizzle below cloud cool and moisten the sub-cloud layer that decrease buoyancy and TKE
(Stevens et al., 1998). As a result, the reduced precipitation increases both TKE and buoyancy
flux in the cloud layer and below cloud (Figure 2e, f).  The enhanced turbulence and buoyancy
support vertical development of clouds, raising cloud tops and expanding the cloud layer upward
(Figures 1b and 2), while also increasing RH near the cloud top (Figure 2d).
On the second day (22 July 2016), the precipitating stratocumulus clouds transition into
non-precipitating thin stratus over the ENA site (Figure S7). The clouds were predominately
overcast from 00-09 UTC and dissipated after 10 UTC, with the domain-mean cloud coverage
decreasing from 0.8-0.9 to 0.1-0.2 (Figure 3a, black line). As shown in Figure S8, the boundary
layer was moist and well-mixed, capped by a sharp temperature inversion, and moisture
decreases rapidly above the inversion. WRF model reproduces the general thermodynamic
structure, including the inversion and moisture decline above the PBL. However, due to biases in
ERA5 thermodynamic profiles, the simulated PBL top is about 700m lower than observed
(Figure S8). Additionally, WRF model fails to capture the rapid decrease of moisture above
cloud top, resulting in a more humid layer above cloud with cloud-top RH of 87% in the model,
compared to 62% in sounding observation (Figure S8c).

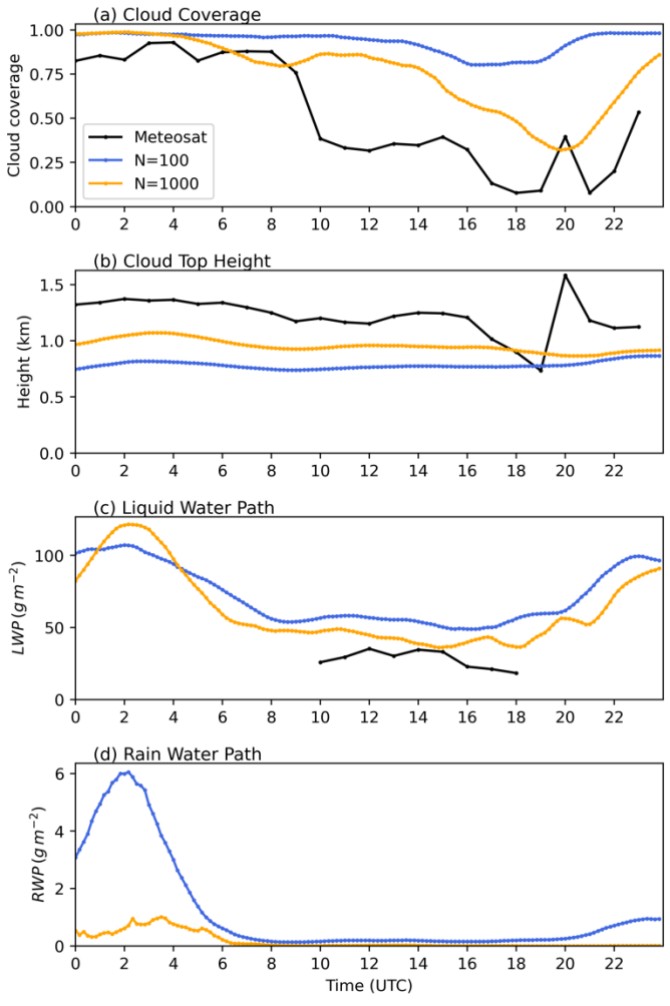

Figure 3. Time series of domain-averaged cloud properties from observations and model
simulation on 22 July 2016. (a) Cloud coverage, (b) cloud top height, (c) cloud liquid water path,
and (d) rain-water path for N=100 (blue lines) and N=1000 (orange lines) experiments.
In the N=100 simulation, the simulated stratocumulus cloud generates light precipitation
from 00-06 UTC, then it transitions to a non-precipitating thin cloud layer after 06 UTC (Figure
3d, blue line). However, the cloud does not dissipate in the model. Domain-mean cloud cover
remains between 0.85 to 0.95 throughout the day, and the simulated LWP is nearly twice that
retrieved from Meteosat (Figure 3a and 3c, blue lines). When aerosol concentrations are
increased to N=1000, clouds dissipate from 14-20 UTC, with a decreasing domain-mean cloud
cover and becoming more consistent with observations (Figure 3a, orange line). Meanwhile,
cloud tops rise slightly with increasing aerosol. The cloud dissipation reflects a net effect of
aerosol induced changes in condensation, evaporation, turbulence, and buoyancy, as shown in
Figure 4.
During the early phase (00–06 UTC), increased aerosol loading suppresses drizzle,
leading to an increase in LWP and a decrease in RWP (Figure 3c, d). Similar as the first case, the
suppressed precipitation enhances turbulence and increases TKE in and below cloud (Figure 4e),
lift the cloud top, and lead to an increase in RH near cloud top (Figure 4d). Meanwhile, the free
tropospheric air above cloud top is relatively drier compared to the first case (Figure 4a). The
increased turbulence and raised cloud top entrain dry air into the cloud and enhances
evaporation. After 6 UTC, as clouds become non-precipitating in the N=100 experiment, the
decrease of cloud water from evaporation starts to dominate the increase from precipitation
suppression and lead to a net decrease in LWP. Reduced buoyancy weakens the upward transport
of moisture and energy from the sub-cloud layer, further contributing to cloud dissipation. As a
result, both cloud cover and LWP decrease with increasing aerosol. (Figure 3a, c).

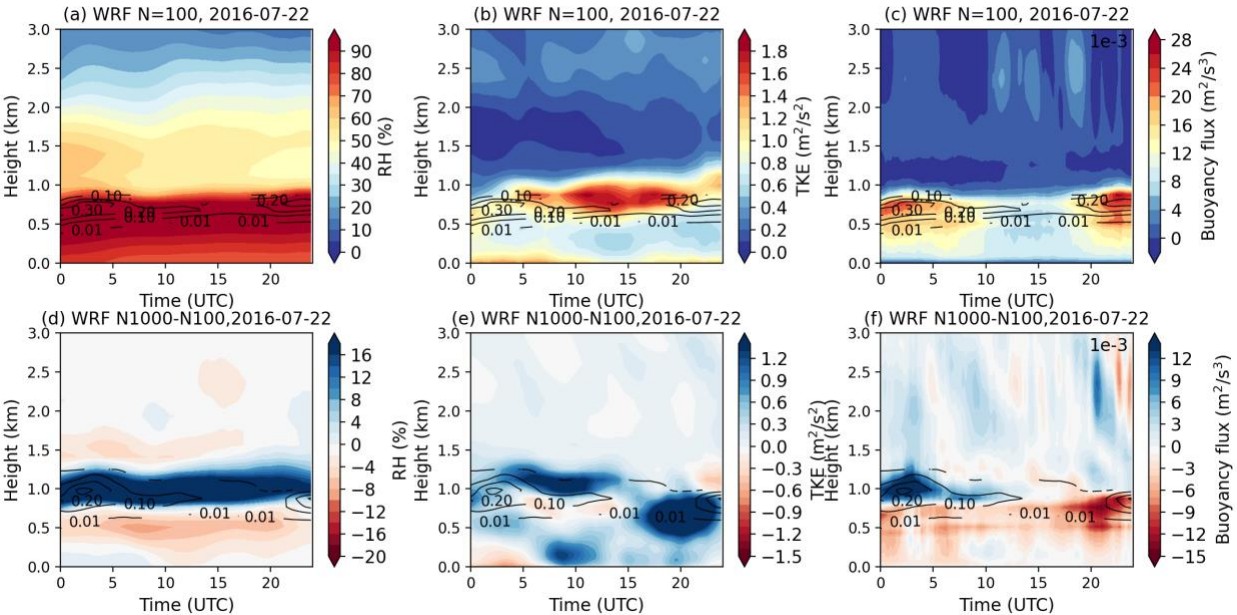

Figure 4. Time series of domain-averaged thermodynamic profiles on 22 July 2016, for (a)
relative humidity, (b) turbulent kinetic energy (TKE) (unit: $m^2 s^{-2}$), (c) buoyancy flux (unit:
$m^2 s^{-3}$) in N=100 simulations, (d) changes in relative humidity profiles, (e) changes in TKE, (f)
changes in buoyancy flux between N=100 and N=1000 simulations. The black contours are
cloud water mixing ratio (unit: g/kg) in (a)-(c) N=100 and (d)-(f) N=1000 simulations.
The absence of afternoon cloud dissipation in WRF simulations are likely associated with
model biases in the thermodynamic structure inherited from ERA5. For example, on 21 July 2016,
ARM sounding observations show a pronounced decrease in specific humidity and relative
humidity above the PBL between 14 and 20 UTC (figures not shown). This sharp drying leads to
cloud erosion in the observations. However, WRF simulations or ERA5 reanalysis produces only a
gradual reduction in moisture from 00 to 20 UTC (Figure 2a), maintaining a moist layer above cloud
top and prevent cloud breakup.  On 22 July 2016, the model reproduces the moisture gradient above
PBL with a warm and dry layer above, the lifted cloud top in the N=1000 simulation entrain dry air
into cloud system and dissipate clouds in the afternoon (Figure 3a). On days when ERA5 accurately
capture the observed moisture decrease above PBL (e.g., 25 and 28 July 2016), the model
reproduces both the dissipation and evening redevelopment of clouds seen in Meteosat data
(figures not shown). This indicates that the diurnal evolution of MBL clouds is highly sensitive
to the representation of diurnal variation in moisture as well as the moisture gradients near the
inversion.
The prescribed, vertically uniform aerosol concentration further reinforces cloud
persistence by maintaining elevated CCN levels and suppressing drizzle formation. The lack of
precipitation scavenging prevents cloud-base evaporative cooling and inhibits decoupling, both
of which would otherwise promote afternoon cloud breakup. The implications of thermodynamic
biases (e.g. the moist layer above cloud top and the underestimated PBL height) for the estimated
ACI are discussed in detail in Section 3.3.2
In a nutshell, precipitating and non-precipitating clouds react differently to aerosol
perturbations in our simulations. For precipitating clouds, aerosols increase LWP through
precipitation suppression and support vertical development of cloud through the impact of
precipitation on PBL dynamic and thermodynamics. For the non-precipitating case, PBL air is
drier compared to the first case, the enhanced turbulence and entrainment of dry air above leads
to evaporation and reduced buoyancy. The reduced buoyancy stabilizes PBL and decays the
cloud layer.
**3.2 Evaluation of LWP Susceptibility Across Cloud States and Synoptic Conditions**
The two cases in Section 3.1 demonstrate the impact of different cloud states and PBL
thermodynamics on cloud responses to aerosol perturbations. In order to evaluate the ACI
process across all simulated cloud states, we composite the cloud fields from all 11 cases and all
three aerosol concentrations (e.g. N=1000 vs. N=100, N=500 vs. N=100, and N=1000 vs.
N=500) to estimate the mean LWP response, and compare it with satellite retrievals, as shown in
Figure 5. More specifically, LWP susceptibility in WRF simulations is defined as the change in
domain mean cloud properties as $dln(LWP)/dln(N_d)$ between polluted and clean simulations
for each 10-minutely model output. To be consistent with satellite retrievals, we focus on
daytime with solar zenith angle less than $65°$. Lastly, we use the LWP-$N_d$ parameter space to
represent different cloud states. (Qiu et al., 2024).
Based on the relationships between $r_e$, LWP, and $N_d$ in the satellite retrievals (e.g.,
$LWP = \frac{4r_e\tau}{3Q_{ext}}$, $N_d = \frac{\sqrt{5}}{2\pi k}(\frac{f_{ad}c_w\tau}{Q_{ext}\rho_w r_e^5})^{1/2}$), $r_e = 15$ isolines is marked in the LWP-$N_d$ parameter
space as an commonly used indicator of precipitation likelihood in the satellite retrieval (e.g.,
Gryspeerdt et al., 2019; Toll et al., 2019; Zhang et al., 2022; Qiu et al., 2024). Based on the
distinct LWP, cloud albedo and CF susceptibilities between cloud states, MBL clouds are
classified into three states: the precipitating clouds ($r_e > 15\ \mu m$), the non-precipitating thick
clouds ($r_e < 15\ \mu m$, LWP$> 75\ gm^{-2}$), and the non-precipitating thin clouds ($r_e < 15\ \mu m$, LWP$<$
$75\ gm^{-2}$) (Qiu et al., 2024). To be consistent with observational reference, the WRF simulated
cloud states are classified using the same definition. Similar to warm MBL clouds in
observations (e.g. Qiu et al., 2024), LWP responses to aerosol perturbation in model simulations
show clear dependence on cloud state (Figure 5a).

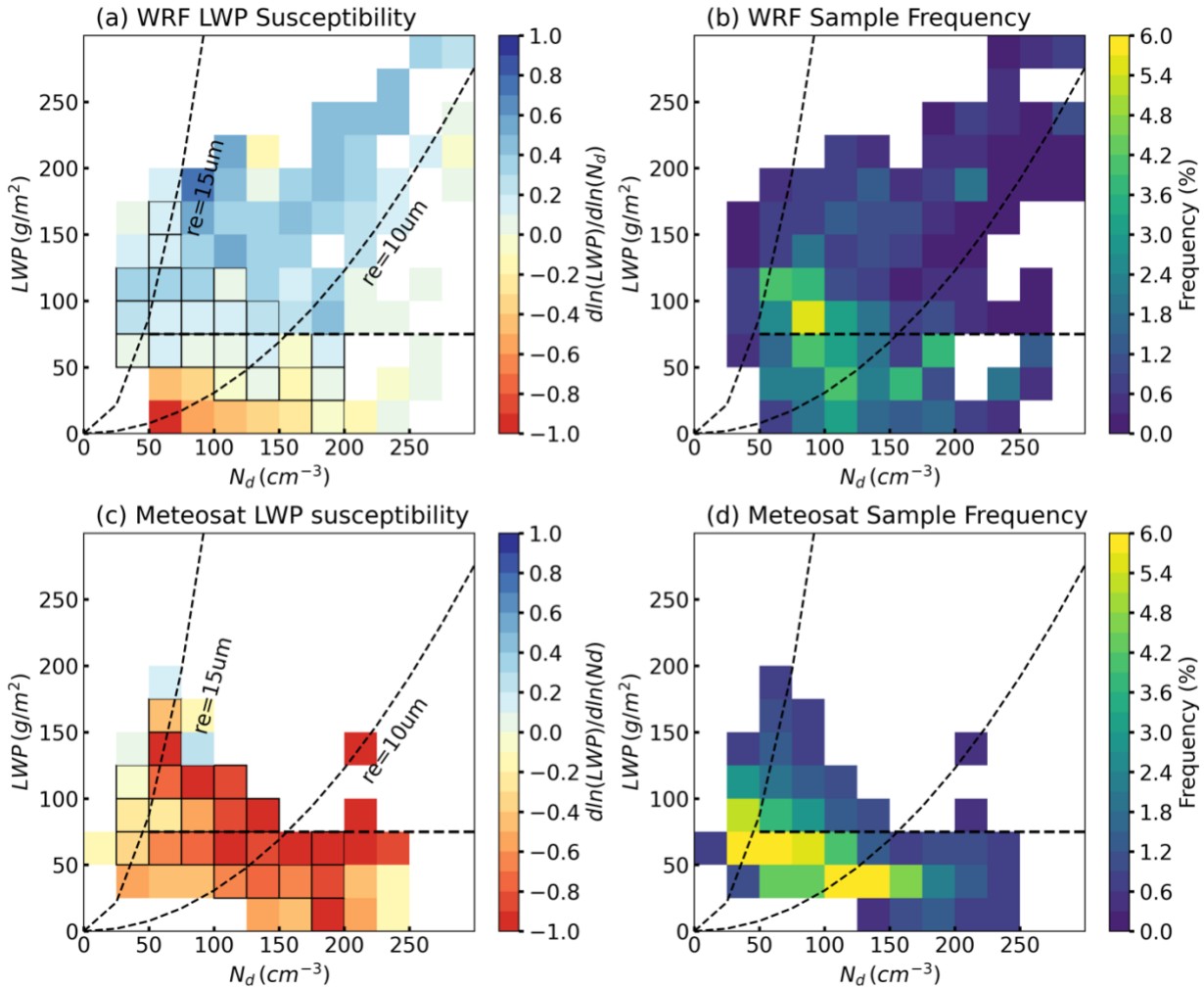

Figure 5. Mean liquid water path (LWP) susceptibility from (a) (b) WRF simulations and (c) (d)
Meteosat cloud retrievals during the daytime. (a) (c) cloud LWP susceptibility $dln(LWP)/$
$dln(N_d)$, (b) (d) frequency of occurrence of sample in each bin. The dashed lines indicate $r_e = 15$
$\mu m$, $r_e = 10 \mu m$, and LWP= 75 $gm^{-2}$, as $r_e$ thresholds for precipitation (precipitating clouds
located to the left of the line), and for thick clouds (with LWP > 75 $gm^{-2}$), respectively. Black-
outlined bins denote cases where the WRF and Meteosat LWP susceptibilities differ significantly
($p < 0.05$) based on a Welch's t-test.
For precipitating clouds ($r_e > 15 \mu m$), LWP slightly increases with $N_d$, with a mean
susceptibility of +0.15. The increase of LWP agrees with the precipitation suppression
mechanism. Meanwhile, there are only 4% of clouds in model simulations locate to the left of the
$r_e = 15 \mu m$ isotherm with small $N_d$, even with aerosol concentration set to 100 $cm^{-3}$ (Figure 5b).
The non-precipitating thick clouds ($r_e < 15 \mu m$, LWP> 75 $gm^{-2}$) is the dominant cloud state in
model simulation, with a total frequency of occurrence of 49%. Different from the evaporation-
entrainment feedback mechanism, LWP largely increases in the model with increasing aerosols,
with a mean susceptibility of +0.32. For non-precipitating thin clouds ($r_e < 15 \mu m$, LWP< 75
$gm^{-2}$), LWP decreases with aerosol perturbations with a mean of −0.14, which is consistent
with the second case shown in the previous section.

To evaluate model performance, we estimated LWP susceptibility from satellite retrievals
within the same domain and for the same 11 cases as the model simulations (Figures 5 c, d).
Specifically, LWP susceptibility was quantified as the regression slope between LWP and $N_d$
within the $1° \times 1°$ domain at each time step of satellite observations. For precipitating clouds,
LWP slightly decreases with increasing $N_d$ in satellite data, consistent with the four-year
climatological mean feature in the ENA region reported in our previous study (Qiu et al., 2024).
This decrease of LWP with increasing $N_d$ is likely associated with the depletion of LWP through
sedimentation–evaporation–entrainment feedbacks, which outweigh the increase of LWP from
precipitation suppression. In contrast, in model simulations, the lack of realistic evaporation-
entrainment feedback results in LWP increasing primarily through precipitation suppression. The
simulated LWP susceptibilities are significantly different with satellite observations at 95%
confidence level for most precipitating clouds (Figure 5a).
For non-precipitating thin clouds, the simulated decrease in LWP with increasing aerosol
concentration agrees in sign with satellite observations. However, the magnitude of this decrease
is weaker, and the simulated susceptibilities remain significantly different from satellite
estimates at 95% confidence level for most bins (Figure 5a, c). This model behavior contrast
with most GCM and coarse CPM studies, which often simulate an increase of LWP for non-
precipitating clouds (e.g., Fons et al., 2024; Christensen et al, 2024; Mülmenstädt et al., 2024;).
The improved representation in our high-resolution simulations arise from better-resolve PBL
turbulence and thermodynamics, which enhance the entrainment of dry air, accelerates
evaporation, reduces buoyancy, and promotes dissipation of the cloud system.
In contrast, for non-precipitating thick clouds, the model and observations diverge
substantially. In satellite observations, LWP decreases most strongly for this cloud state, with a
mean LWP susceptibility of $-0.69$ (Figure 5c). This observational estimate is consistent with the
climatological mean derived from four years of Meteosat data over the ENA region (Qiu et al.,
2024). In the model, however, LWP increases most strongly with increasing $N_d$ for this cloud
state. Moreover, compared with satellite retrievals, model simulates substantially larger
population of polluted thick clouds characterized by high $N_d$ and LWP. For example, non-
precipitating thick clouds are the dominant cloud state in the model, accounting for 49% of total
cloud occurrence (Figure 5b), whereas they are the least frequent in observations, at only 15.7%
(Figure 5d). Meanwhile, only 4% of simulated clouds fall into the precipitating cloud regime
with $N_d < 50$, compared to a 22.2% in the satellite observations
The overall overestimation of $N_d$ likely arises from the prescribed aerosol concentration
in the model configuration, combined with the absence of precipitation scavenging. For
reference, the mean aerosol concentration over the ENA region during summer is approximately
400 $cm^{-3}$ (e.g., Zhang et al., 2021; Wang et al., 2021; Zheng et al., 2024). The model's
overestimation of LWP may stem from its excessively positive LWP susceptibility in thick
clouds. As shown in Figure S9, simulated LWP in the N=100 experiment agrees reasonably well
with the Meteosat retrievals, with a mean value about 10% lower than observed. However, in the
N=500 and N=1000 simulations, the strong positive LWP susceptibility leads to increases in
LWP for clouds with LWP> 75 $gm^{-2}$, resulting in mean values 30% and 40% higher than
Meteosat retrievals, respectively.
To further examine whether these discrepancies depend on large-scale meteorological
conditions, we assessed LWP susceptibility across different synoptic regimes. Because only one
case is available for the "weak-trough" regime (Table S1), our comparison focuses on the "high-
ridge" and the "post-trough" regimes (Figure S10). The "high-ridge" regime shows a higher
occurrence of non-precipitating thin clouds than the "post-trough" regime, with total frequencies
of 49% and 40%, respectively (Figures S10b, d). This more frequent non-precipitating thin cloud
in the model is consistent with our previous study based on six years of ground-based
observations at the ARM ENA site, which revealed that the "high-ridge" regime favors single-
layer stratocumulus clouds with shallower cloud depth and smaller LWP compared to the "post-
trough" regime (Zheng et al., 2025).
In addition, non-precipitating thin clouds in the "high-ridge" regime exhibit more
negative LWP susceptibilities than clouds with similar LWP and $N_d$ in the "post-trough" regime.
This difference in LWP susceptibility is associated with the colder and drier air above clouds
under subsidence in the "high-ridge" regime, which facilitates cloud dissipation, as also
demonstrated in the case study. Furthermore, non-precipitating or lightly drizzling thick clouds
in both synoptic regimes manifest strong positive LWP susceptibilities, suggesting that the
model-observation discrepancy for this cloud state persist regardless of synoptic conditions and
therefore warrants further investigation. In summary, the mean LWP susceptibility from our
simulations were evaluated against satellite retrievals in the LWP-$N_d$ parameter space across
different cloud states and synoptic conditions for a comprehensive comparison. The simulations
reproduce the observed decrease in LWP for non-precipitating thin clouds, although with weaker
magnitudes. For precipitating clouds, the model predicts a slight increase in LWP instead of the
weak decrease seen in satellite observations, reflecting the limited representation of evaporation-
entrainment feedback in the model. Large discrepancies remain for non-precipitating or lightly
drizzling thick clouds, where the model simulates too many polluted thick clouds and yields an
opposite (positive) LWP response compared to the strongly negative satellite signal.
In addition, the model-observation discrepancy persists across all synoptic regimes,
suggesting that they originate from the model's representation of cloud microphysics,
precipitation, and aerosol-cloud coupling rather than from large-scale meteorological variability.
The consistency of these modeled LWP response, in agreement with previous LES studies of
similar cloud regimes (e.g., Wang et al., 2020; Lee et al., 2025), further motives the central focus
of the next section: diagnosing the physical mechanisms driving these biases. We show that three
leading factors dominate the discrepancy: excessive precipitation production in thick clouds, a
moist bias above cloud top, and satellite retrieved $N_d$-LWP relationships contaminated by
internal cloud processes.

**3.3 Causes of Satellite–Model Discrepancies in LWP Susceptibility**

The satellite–model differences highlighted above point to systematic biases in how the
model represents cloud microphysics, precipitation processes, and entrainment pathways. In this
section, we diagnose the physical mechanisms driving these discrepancies, beginning with the
model's precipitation efficiency.

**3.3.1 Precipitation Efficiency**

A long-standing challenge in numerical models is the tendency to produce precipitation
too frequently and too lightly (Sun et al., 2006; Stephens et al., 2010). To assess the modeled
precipitation efficiency with observation, Figure 6 shows the mean cloud properties from
Meteosat observations, and from WRF simulations for the 11 cases combining all three aerosol
concentrations (N=100, 500, and 1000). As satellite retrieves $r_e$ near cloud top, we used $r_e$ at
~100 m below cloud top in the simulation, which approximate $\tau = 2$ from cloud top for marine
stratocumulus. The modeled pixel-level precipitation fraction is calculated as the area fraction of
cloudy pixels with the column maximum radar reflectivity ($Z\ max$) greater than $-15$ dBZ at
each model output time (Haynes et al., 2009; Suzuki et al., 2015; Jing et al., 2017). Modeled
radar reflectivity is from the radar simulator (CR-SIM), as discussed in the methodology. The
precipitation fraction in Meteosat is calculated as the area fraction of clouds with $r_e > 15\ \mu m$.
Qiu et al. (2024) evaluated different effective radius thresholds and rain rate thresholds in
satellite retrievals using precipitation masks derived from ground-based radar reflectivity at the
ENA site, and concluded that the $r_e > 15\ \mu m$ threshold showed the best agreement with
observations.

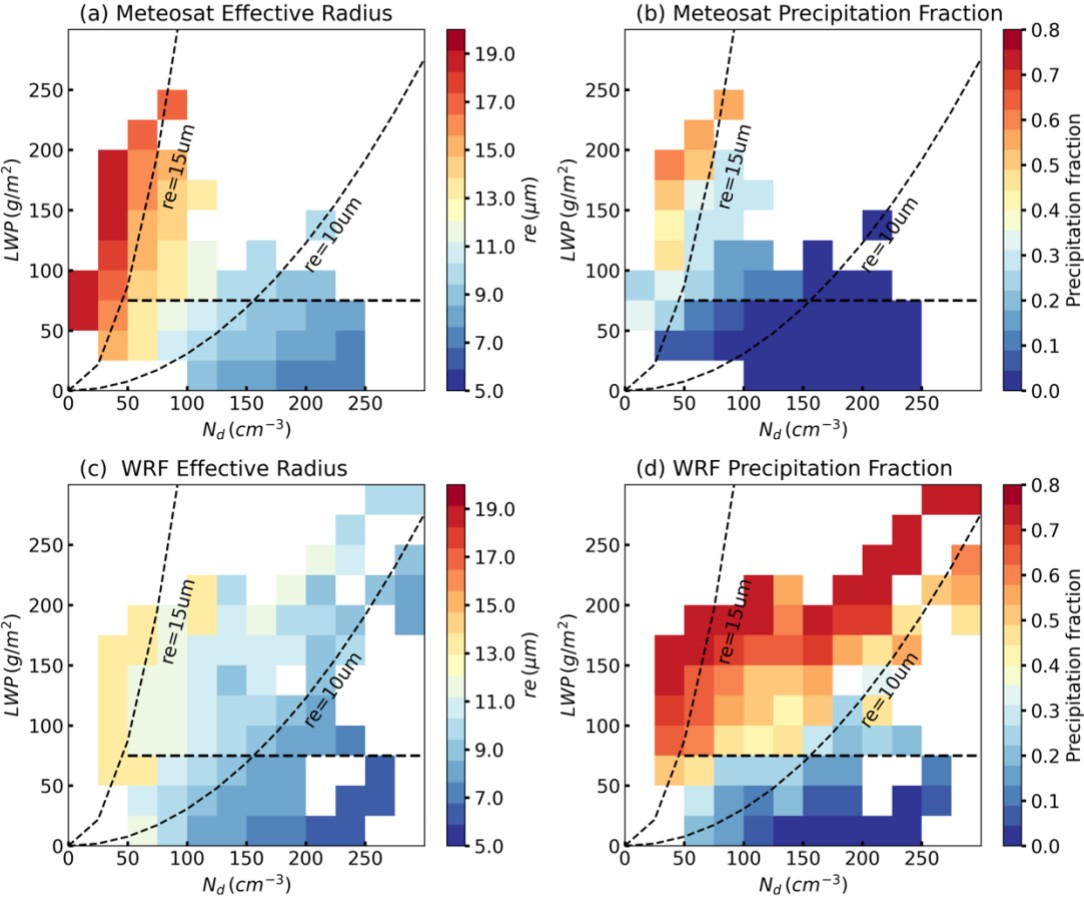

Figure 6. Mean cloud properties from (a), (b) Meteosat retrievals and (c), (d) WRF simulations
during the daytime. (a), (c) effective radius, (b), (d) pixel-level precipitation fraction. The dashed
lines indicate $r_e$ =15 $\mu m$, $r_e$ =10 $\mu m$, and LWP= 75 $gm^{-2}$, as $r_e$ thresholds for precipitation
(precipitating clouds located to the left of the line), and for thick clouds (with LWP > 75 $gm^{-2}$),
respectively.
As shown in Figures 6a, c, the modeled $r_e$ is ~1-3 $\mu m$ smaller than satellite retrievals for
a similar cloud condition. Additionally, compared to observation, model generates precipitation
too often at smaller drop size with $r_e > 10\ \mu m$ and at higher $N_d$ concentration (Figures 6b, d,
$r_e$ =10 $\mu m$ dashed line). The large discrepancy in LWP susceptibility for thick clouds between
the 10 and 15 $\mu m$ isolines is likely linked to model bias in precipitation efficiency. To further
investigate the model bias of excessive rain at smaller drop size and the positive LWP responses
to aerosol perturbations, we compared the modeled radar reflectivity profiles from the radar
simulator with ARM observations using the CFODD framework. Based on the relationship
between $Z_e$ and the droplet collection efficiency ($E_c$), the vertical slope of $Z_e$ as a function of in-
cloud optical depth ($\tau_d$) is directly linked to $E_c$, a steeper slope indicates a larger $E_c$ (Suzuki et
al., 2010).

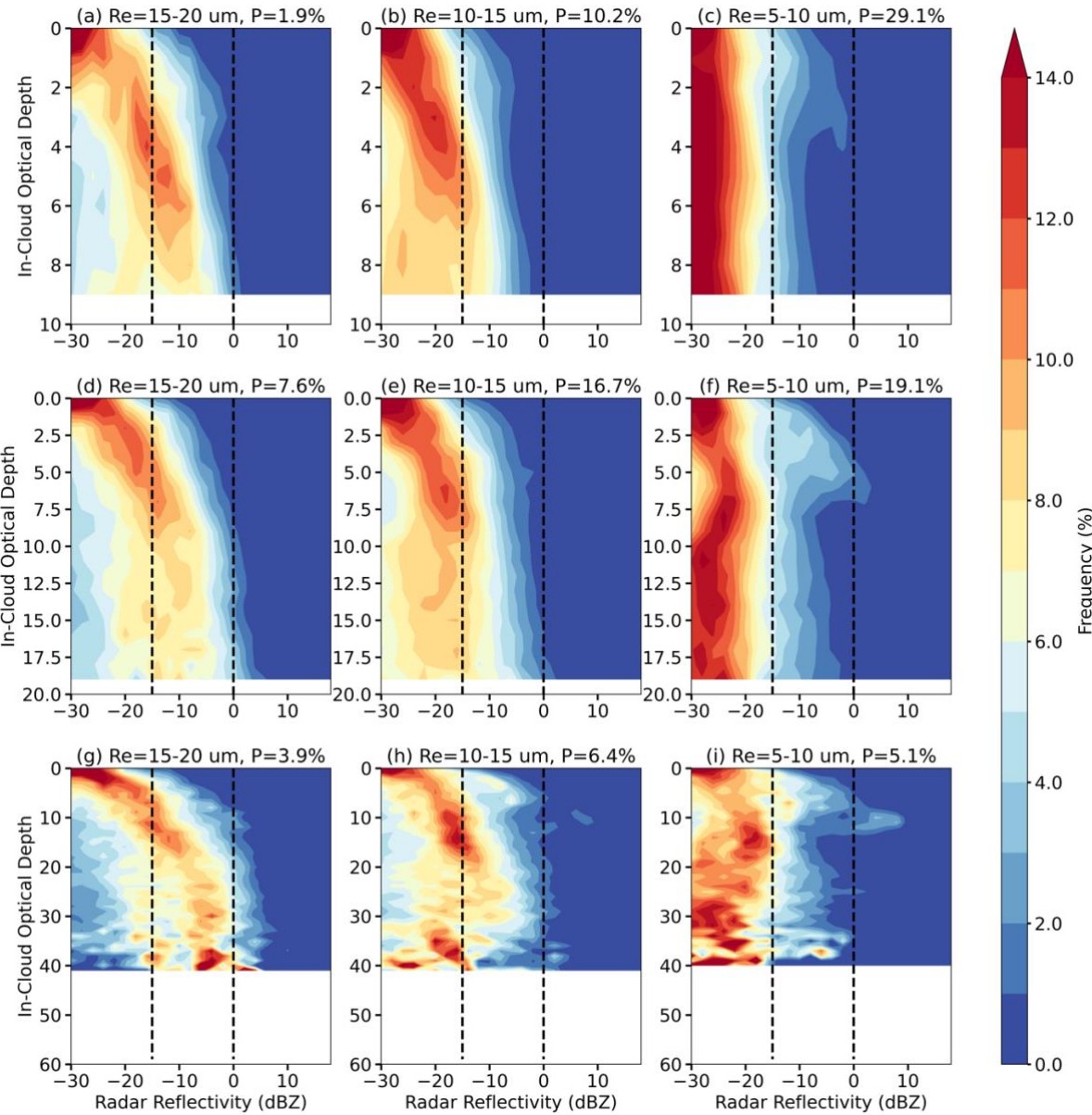

Figure 7. Frequency of radar reflectivity as a function of in-cloud optical depth ($\tau_d$) for ARM
ground-based observations during the daytime. Different rows are for different ranges of optical
depth ($\tau$): (a)-(c) clouds with $\tau < 10$, (d)-(f) clouds with $10 < \tau < 20$, (g)-(i) clouds with $\tau > 20$.
Different columns are for different ranges of effective radius ($r_e$). The left, middle, and right
columns are for $15 - 20\ \mu m$, $10 - 15\ \mu m$, and $5 - 10\ \mu m$, respectively. The black dashed lines
in each panel denote $-15$ dBZ and $0$ dBZ, as thresholds of drizzle and rain, respectively. The
percentage of sample (P) for each subgroup is denoted in the figure, with a total sample of
607 91,737.

Ground-based radar reflectivity profiles and cloud retrievals at the ARM ENA site are
used as the ground truth. To reduce noise, radar reflectivity profiles and cloud boundary data are
smoothed to a 1-minute resolution. To increase the sample size, we analyzed the climate-mean
radar reflectivity profiles of stratocumulus and cumulus clouds observed during the summer
months (June to August) from 2016 to 2021, comprising a total of 91,737 profiles. Radar
reflectivity profiles derived from the selected 11 cases exhibit consistent characteristics (figure
not shown). To better distinguish microphysical processes such as autoconversion and accretion
from dynamical processes such as updraft, clouds are further categorized by both $r_e$ and $\tau$ ranges.
MBL clouds are classified as non-precipitating clouds, drizzle, and rain using a reflectivity
threshold of $Z_e$ < -15 dBZ, -15 dBZ < $Z_e$ < 0 dBZ, and $Z_e$ > 0 dBZ, respectively, as denoted
by black dashed lines in Figures 7 (Haynes et al., 2009; Suzuki et al., 2015; Jing et al., 2017).
Applying the same cloud state classification as in the satellite observations (e.g., $r_e$ >
15 $\mu m$ for precipitating clouds and LWP > 75 $gm^{-2}$ for thick clouds), the total frequency of
occurrence of precipitating, non-precipitating thin, and non-precipitating thick clouds are 30.7%,
46.3%, and 23.0%, based on six-year of ARM observations. These frequencies are consistent
with those derived from satellite data for the 11 cases (22.2%, 55.6%, and 22.2%, respectively;
Figure 5d). Therefore, the selected cases in this study are representative of the typical
distribution of MBL cloud types in the ENA region during summer.
As shown in the first column of Figure 7, in clean environment with $r_e$ > 15 μm, the
observed MBL clouds start to drizzle with $Z_e$ > −15 dBZ even in the thinnest category (Figure
7a), of which the cloud top is mostly non-precipitating ($Z_e$ < −25 dBZ). Cloud drops rapidly
grow from cloud top downward and initiate drizzle at ~ 4-6 optical depth into the cloud.
However, most observed MBL clouds, even for the thickest category (Figure 7g), remain
drizzling rather than raining as most of the radar reflectivity is lower than 0 dBZ.
Figures 7b, e, h represent clouds with observed $r_e$ of $10 - 15 \ \mu m$, indicating an increase
in $N_d$ compared with clouds with similar $\tau$ and $r_e$ > 15 um (Figures 7a, d, g). Precipitation in
these clouds is suppressed as the $Z_e$ is mostly less than –15 dBZ in thin clouds ($\tau$ < 10, Figures
7b). Thick clouds produce drizzle at ~$\tau_d$ > 20 and $Z_e$ slightly decrease at cloud base, likely due
to mixing and evaporation (Figure 7h). When $r_e$ decreases to below 10 $\mu m$ (Figures 7c, f, i), $Z_e$
further reduces to around –20 to –30 dBZ throughout the cloud layer, indicating that precipitation
is further suppressed. The precipitation suppression effect is shown not only by the peak
frequency of $Z_e$, but also the slope of $Z_e$, which indicates the droplet collection efficiency as
discussed above. As seen in Figure 7, for clouds with similar thickness, the slope of $Z_e$ decreases
with decreasing $r_e$, which reflects a weaker collision coalescence and accretion processes with
higher $N_d$ and smaller cloud drops.
In thick clouds with $r_e < 10 \ \mu m$ (Figure 7i), most radar reflectivity remains below –25
dBZ in the lower cloud layer, while reflectivity slightly increases toward cloud top in the region
corresponding to ~10-20 optical depth into the cloud. Reflectivity then decreases again toward
cloud top. This vertical pattern is consistent with the structure of marine clouds reported in
Suzuki et al. (2010). The observed decrease of reflectivity near cloud top may be attributed to
entrainment and evaporation, or to the accretion process involving large droplets falling
downward, as indicated by localized reflectivity peaks exceeding -15 dBZ (Figure 7i).
Meanwhile, in clouds with small drop sizes, cloud deepening or dynamical processes have little
effect on precipitation based on observations.

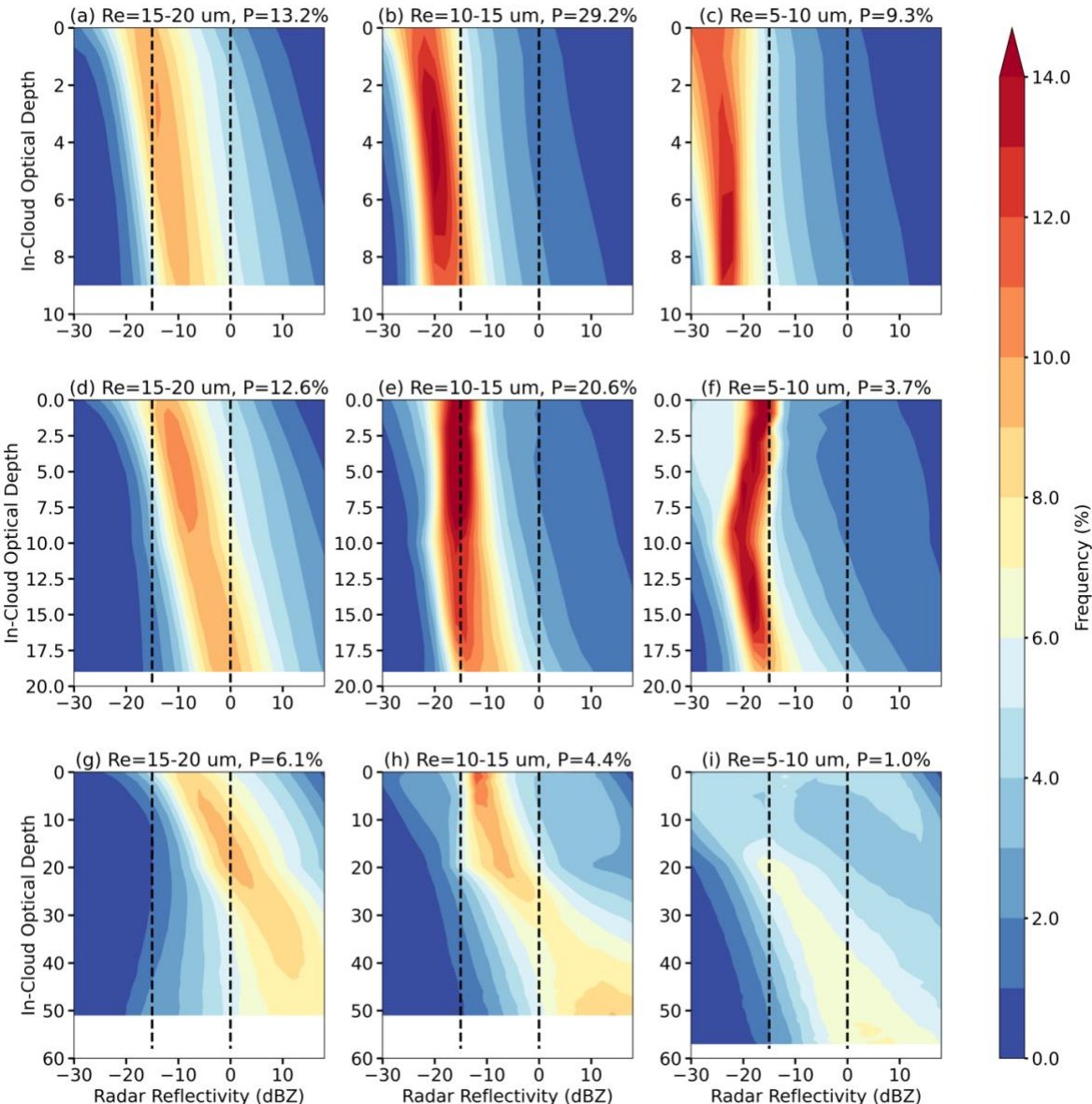

Figure 8. Frequency of radar reflectivity as a function of in-cloud optical depth ($\tau_d$) for WRF
N=100 simulation. Different rows are for different ranges of optical depth ($\tau$): (a)-(c) clouds with
$\tau < 10$, (d)-(f) clouds with $10 < \tau < 20$, (g)-(i) clouds with $\tau > 20$. Different columns are for
different ranges of effective radius ($r_e$). The left, middle, and right columns are for $15 - 20\ \mu m$,
$10 - 15\ \mu m$, and $5 - 10\ \mu m$, respectively. The black dashed lines in each panel denote $-15$
dBZ and 0 dBZ, as thresholds of drizzle and rain, respectively. The percentage of sample (P) for
each subgroup is denoted in the figure.
Compared to the "ground truth", our model simulations reasonably identify the non-
precipitating regime in clouds with $r_e < 10\ \mu m$ and $\tau < 20$, when cloud drops are too small for
efficient collision coalescence (Figures 8c, f). Additionally, drizzle initiates at the same $r_e$ and $\tau$
ranges as in observations: for example, the maximum frequency of $Z_e$ exceeds $-15$ dBZ in thin
clouds with $r_e > 15\ \mu m$ and $\tau < 10$ (Figure 8a) or in thick clouds with $r_e = 10 - 15\ \mu m$ and $\tau =$
$10 - 20$ (Figure 8e). This result is different from GCM or GCPM where models hardly simulate
any non-precipitating clouds, or drizzle initiate too early in the cloud (e.g. 5-10 optical depth;
Jing et al. 2017, 2019; Michibata and Suzuki, 2020). The better resolved non-precipitating
regime as well as the transition from non-precipitating cloud to drizzle process in our simulations
reveal the importance of model resolution to better simulate precipitation.
On the other hand, model overestimates precipitation in both intensity and the frequency
of occurrence in optical thick with $\tau > 20$, the simulations produce rain with peak $Z_e$ exceeding 0
dBZ in all size ranges, even in clouds with $r_e < 10\ \mu m$ (Figure 8 g-i). Furthermore, precipitation
initiates too early near cloud top: all precipitating clouds in the model start to drizzle or even rain
at cloud top (Figures 8a, d, e, g, h, i). Based on the features shown in the CFODD analysis, the
overestimation of precipitation could be attributed to the following four aspects in the
parameterization.
First, the overestimation of reflectivity at cloud top indicates that autoconversion is
activated too early in clouds near the top. With the same aerosol concentration, clouds with less
activated $N_d$ exhibit larger $r_e$ (Figure 6c). As the autoconversion rate scaled non-linearly with $N_d$
(e.g. $\frac{\partial q_c}{\partial t} = 1350 q_c^{2.47} N_d^{-1.79}$), clouds with larger drop size (e.g. ~15–20 $\mu m$) have smaller $N_d$,
and therefore exhibit larger autoconversion rate. Second, the overestimation of reflectivity near
cloud top could be due to underestimation of entrainment rate or evaporation rate from the moist
layer above. As seen in Figure 8, the simulated $Z_e$ does not decrease towards cloud top or cloud
base as in the observations, which indicates an underestimation of entrainment and evaporation.
Third, the overproduction of rain in the model indicates an overestimation of the accretion
process. In the Morrison scheme, accretion is parameterized as a function of cloud water and
rainwater content; thus, when autoconversion is triggered too early, accretion also initiates too
early. This bias is amplified in thick clouds, which have greater liquid water content and longer
path for droplet collection (Figures 8 g-i). For thick clouds with small drop size (Figure 8i), they
remain non-precipitating at the cloud top, indicating that autoconversion is appropriately
suppressed by small drop size. However, these clouds still produce rain, suggesting an
overestimation of accretion. Lastly, the excessive rain production in thick clouds also point to an
overly broad parameterized drop size distribution (DSD), which lead to an early initiation of
autoconversion at cloud top and rain formation in clouds with large $r_e$.
Overall, in N=100 simulation (Figure 8), most modeled MBL clouds are optically thin ($\tau$
$< 20$) and exhibit medium ($r_e = 10 - 15\ \mu m$, 49.8%) or large droplet sizes ($r_e = 15 - 20\ \mu m$,
25.8%). Compared to observations, model produces more clouds with larger drop size, while
observations show a majority with $r_e < 10\ \mu m$ (53.3%; Figure 7, third column). Meanwhile,
although the aerosol concentration is prescribed, the model predicts $N_d$ through aerosol
activation and microphysical processes, resulting in variabilities in $N_d$. For clouds with given
optical depth, a decrease in $r_e$ indicates an increase in $N_d$. This increase in $N_d$ is associated with
both lower peak of $Z_e$ and a reduced vertical $Z_e$ gradients in the CFODD, suggesting aerosol-
induced precipitation suppression. Lastly, cloud dynamics plays a stronger role in the simulation
than in observations. For example, thicker clouds in the model show higher peak $Z_e$ values and
broader $Z_e$ distribution than thinner clouds with same $r_e$, whereas this enhancement is less
evident in ARM observations.

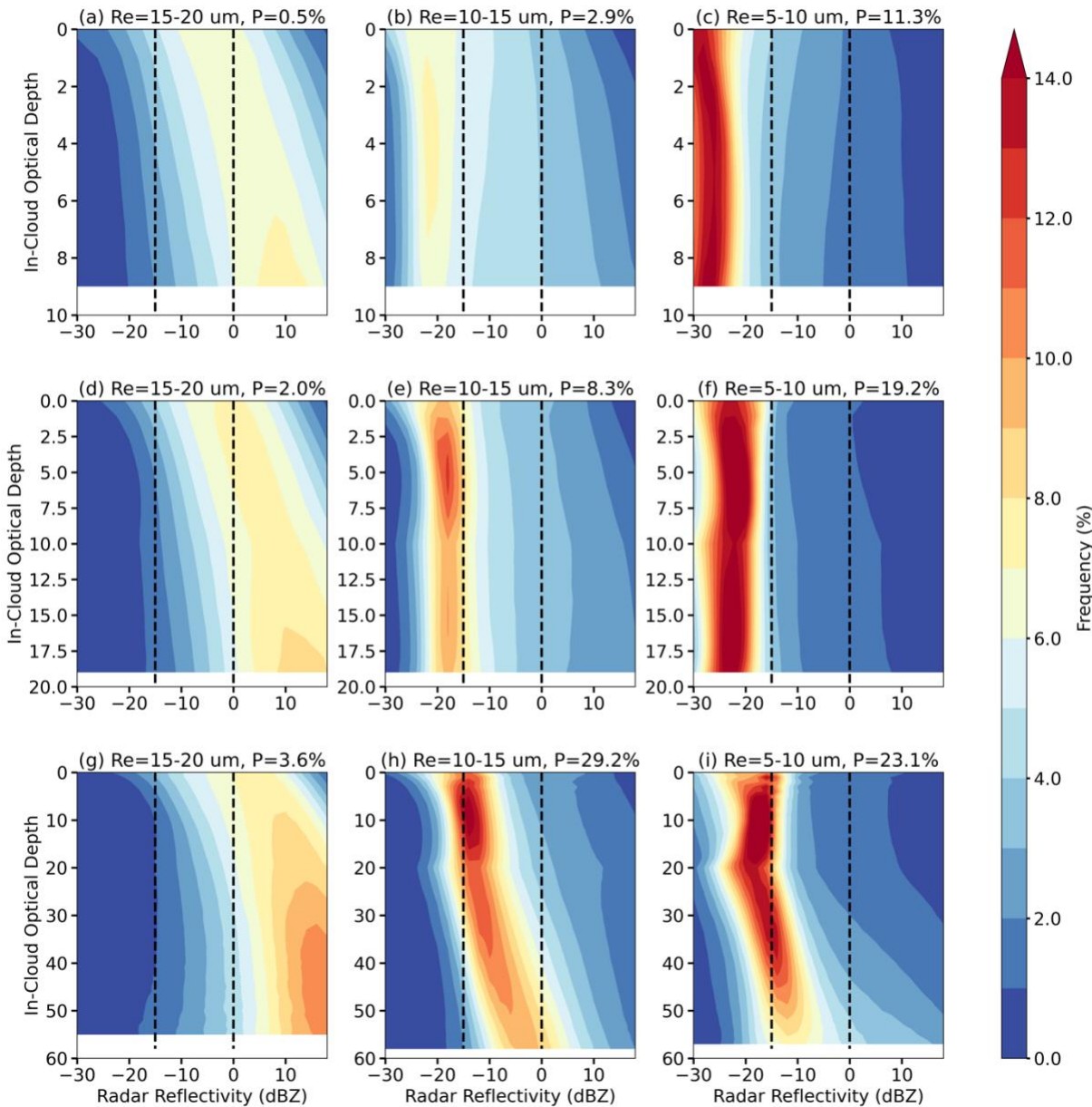

Figure 9. Frequency of radar reflectivity as a function of in-cloud optical depth ($\tau_d$) for WRF N=500 simulation. Different rows are for different ranges of optical depth ($\tau$): (a)-(c) clouds with $\tau < 10$, (d)-(f) clouds with $10 < \tau < 20$, (g)-(i) clouds with $\tau > 20$. Different columns are for different ranges of effective radius ($r_e$). The left, middle, and right columns are for $15 - 20\ \mu m$, $10 - 15\ \mu m$, and $5 - 10\ \mu m$, respectively. The black dashed lines in each panel denote $-15$ dBZ and 0 dBZ, as thresholds of drizzle and rain, respectively. The percentage of sample (P) for each subgroup is denoted in the figure.

Comparing simulations with different prescribed aerosol concentrations, we observe that with increasing aerosols and decreasing drop size, precipitation is suppressed. This is evidenced by the shift of frequency of occurrence of precipitating clouds, along with reduced peak $Z_e$ and shallower gradient of $Z_e$. For example, the most common cloud type shifts from thin clouds with

moderate $r_e$ in the N=100 simulation (Figures 8b, e) to thicker clouds with smaller $r_e$ in the
N=500 run (Figures 9h, i), revealing a typical cloud response to precipitation suppression.
Meanwhile, the percentage of clouds with $r_e = 15 - 20 \ \mu m$ decreases significantly from 31.9%
in N=100 to 6.1% in N=500 simulations. As a result, the droplet size distribution in N=500
simulation aligns better with ARM observations, although clouds are still thicker in model. For
clouds with similar $r_e$ and $\tau$, both the peak $Z_e$ and its vertical gradient decrease with increasing
aerosol concentrations due to the reduced autoconversion with higher $N_d$. In particular, thick
clouds with medium $r_e$ ($r_e = 10 - 15 \ \mu m$, $\tau >$20, Figure 9h) transition from raining to drizzling
in the N=500 simulation, aligning more closely with observations.
For clouds with $r_e > 15 \ \mu m$, rain becomes stronger compared to the N=100 simulation,
even in the thinnest cloud (Figures 9 a, d, g vs. Figures 8 a, d, g). While the enhancement of
precipitation with increasing aerosol concentration may initially seem counter-intuitive, it can be
explained by the parameterization of DSD in the model. For clouds with similar $\tau$, increasing $r_e$
is associated with higher LWP and $q_c$, but lower $N_d$. Based on Equation (5), the slope parameter
$\lambda$ decreases with increasing $r_e$, resulting in a broader DSD with a flatter slope. Additionally, the
dispersion parameter $\eta$ is proportional to $N_d$ so that polluted clouds in N=500 simulation also
exhibit broader DSDs. As a result, even under suppressed autoconversion due to higher $N_d$, the
extended tail of the broader DSD initiates autoconversion, enhances accretion from higher fall
speed, and ultimately enhances precipitation in the N=500 simulation. Note that this type of
cloud occurs much less frequently in the N=500 simulation (6.1%) than in the N=100 simulation
(31.9%).
When continuously increasing aerosol concentration from N=500 to N=1000 (Figure 9
vs. Figure 10), the CFODD of reflectivity changes little, indicating a saturation of the
precipitation suppression effect and the broadening of DSD. More clouds shift to the non-
precipitating thick clouds subgroup with $r_e < 10 \ \mu m$ and $\tau >$20 (44.6%, Figure 10i).
In summary, we evaluated the vertical development of precipitation in the model using
ARM radar reflectivity profiles. Our simulations realistically reproduce the non-precipitating
regime and the transition to drizzling clouds at similar $r_e$ and $\tau$ ranges as ARM observations.
Meanwhile, model overestimates precipitation for optically thick clouds and clouds with $r_e >$
15 $\mu m$. This overestimation could be attributed to the early initiation of the autoconversion
process, which leads to an early onset of rain near the cloud top. The excessive accretion rates,
along with underestimation of entrainment and evaporation, lead to an overproduction of rain in
the model, especially in thick clouds with larger water content and longer droplet collection path.
Additionally, the parameterized DSD is too broad in the model, especially for polluted clouds
with large $N_d$ and large $r_e$.
As the model reasonably captures the properties of non-precipitating thin clouds in
agreement with ARM observations, the simulated LWP susceptibility aligns well with satellite-
based estimates. In contrast, the overestimation of precipitation in thick clouds leads to a
predominantly positive LWP susceptibility in the model due to the precipitation suppression
effect. However, satellite observations indicate that these clouds are typically non-precipitating,
where entrainment drying dominates, resulting in a negative LWP susceptibility. This highlights
the need to improve the parameterization of precipitation processes: particularly autoconversion,
accretion, and DSD representation, in order to better simulate ACI across all cloud regimes.

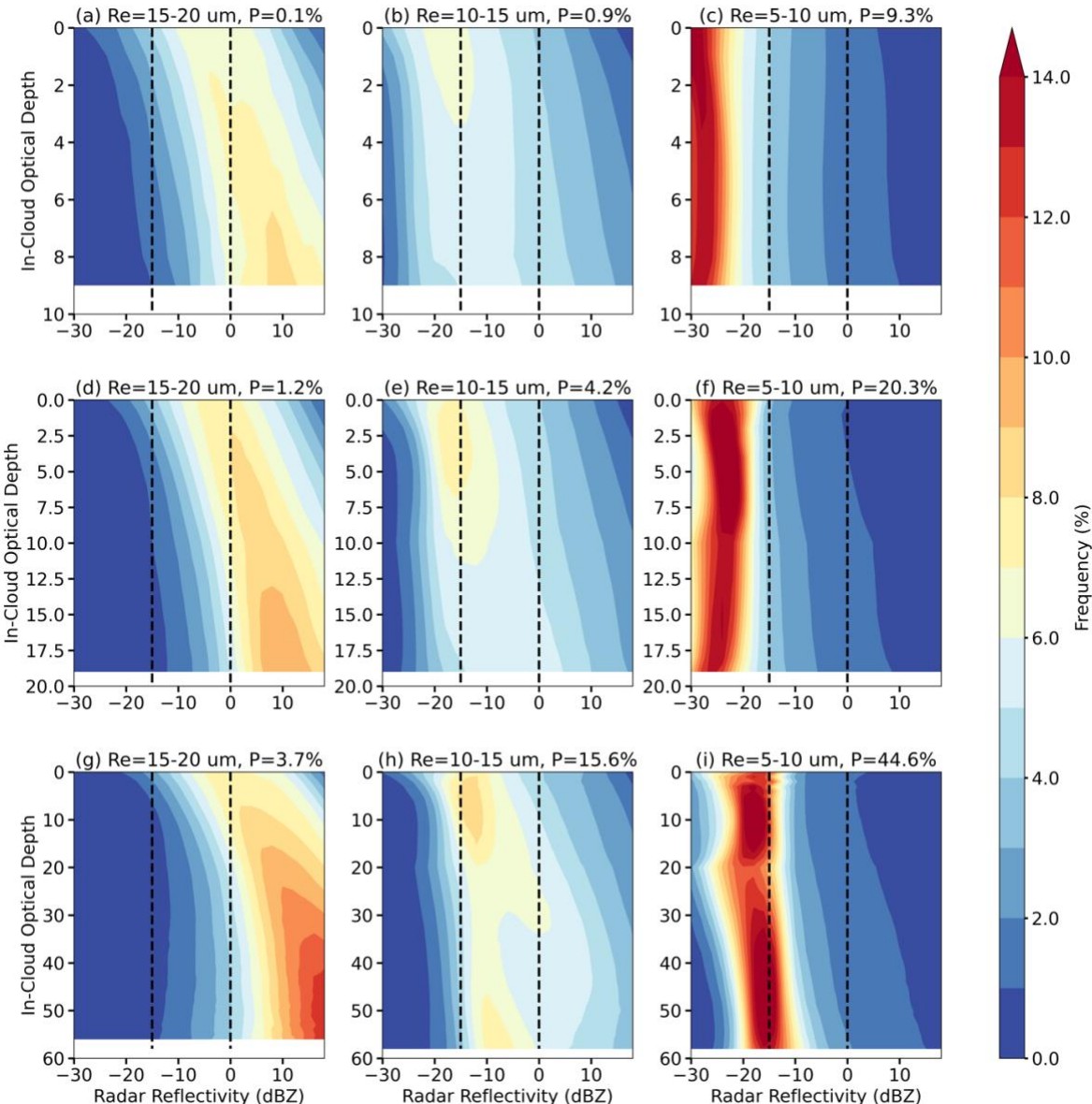

Figure 10. Frequency of radar reflectivity as a function of in-cloud optical depth ($\tau_d$) for WRF N=1000 simulation. Different rows are for different ranges of optical depth ($\tau$): (a)-(c) clouds with $\tau < 10$, (d)-(f) clouds with $10 < \tau < 20$, (g)-(i) clouds with $\tau > 20$. Different columns are for different ranges of effective radius ($r_e$). The left, middle, and right columns are for $15 - 20\ \mu m$, $10 - 15\ \mu m$, and $5 - 10\ \mu m$, respectively. The black dashed lines in each panel denote $-15$ dBZ and 0 dBZ, as thresholds of drizzle and rain, respectively. The percentage of sample (P) for each subgroup is denoted in the figure.

While our analysis focuses on the two-moment Morrison scheme, Christensen et al. (2024) found that the choice of microphysics and PBL schemes accounts for only about 30 % of the variability in simulated ACI, much smaller than the variability across meteorological conditions and cloud states. Since this study encompasses 11 cases spanning diverse synoptic regimes and cloud types, the overall conclusions are unlikely to change substantially with

alternative two-moment bulk microphysics schemes. Nonetheless, future investigations using
multiple microphysics schemes would be valuable for quantifying the robustness of the
precipitation parameterization and its role in ACI uncertainty.

**3.3.2 Model Bias in Capturing Inversions**

As discussed in case study in Section 3.1, ERA5 profiles fail to accurately represent the
location and strength of inversions over the ENA region. These biases lead to an underestimated
boundary layer height and an overestimated RH above cloud top in the simulations. Figure 11
compares the probability density function (PDF) of cloud-top RH between ARM sounding
observations and WRF simulations across all 11 cases for N=1000 simulation. Different aerosol
concentrations (e.g., N=100, N=500) show consistent results (not shown). In ARM observations,
cloud-top height is derived from the radar reflectivity profile, as described in the method section;
while in WRF simulations, cloud top is defined as the highest model level where the cloud water
mixing ratio exceeds 0.001 g/kg. The RH is sampled at ~100m above cloud top in both data. We
further compare the cloud-top heights in WRF simulations defined using cloud water mixing
ratio and radar reflectivity profiles with $Z_e$ > -40 dBZ from the radar simulator. The two
approaches yield nearly identical results, with a mean difference of less than 40m (figure not
shown).

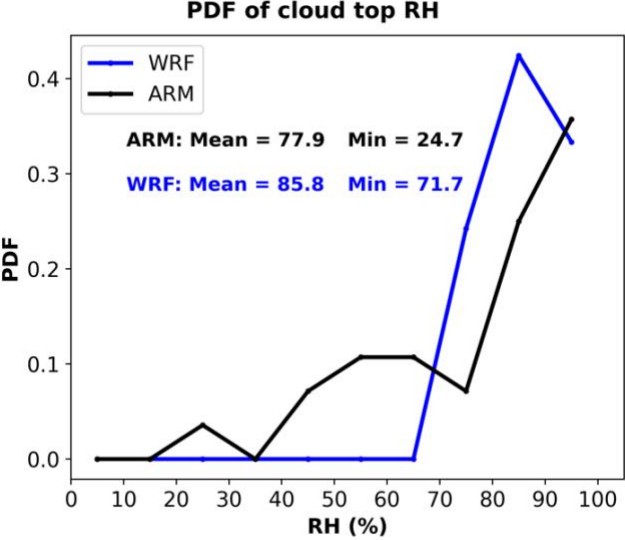

Figure 11. PDF of cloud top relative humidity (RH) for WRF simulations (blue line) and ARM
sounding observations (black line).
To ensure a meaningful comparison between WRF output and ground-based
observations, cloud-top RH from WRF is averaged over a 10km × 10km grid box centered at the
ARM ENA site for each sounding time, given the ~1.2-1.4 km mean cloud-top height for MBL
clouds and ~7 m/s prevailing wind speed at ENA during summer (Wood et al., 2015; Wu et al.,
2020). As seen in Figure 11, WRF simulations exhibit a systematic wet bias in cloud-top RH,
with the mean values 7.9% higher than those from observations and with no RH values below
71%.
Figure 12 shows the mean relationship between clout top RH and cloud susceptibilities
calculated based on domain mean values for all three simulations (e.g. N=1000 vs. N=100,
N=500 vs. N=100, and N=1000 vs. N=500). The cloud top RH is the domain mean RH value at
~100m above cloud top for all simulations. As seen in Figure 12a, we find a positive correlation
between cloud-top RH and LWP susceptibility in the simulations, which is consistent with cloud
responses shown in case study where a dry layer above cloud promotes evaporation and decrease
LWP. Additionally, these positive relationships are consistent among different aerosol
concentrations (e.g., N=1000 vs. N=100 or N=500 vs. N=100; figures not shown). Meanwhile, as
seen in Figure 12, cloud top moisture has a more evident impact on cloud LWP than cloud cover.
Relations between cloud top moisture and cloud susceptibilities found in our simulations are
consistent with that in satellite observations around the globe (e.g. Toll et al., 2019; Yuan et al.,
2023), except that LWP susceptibility is mostly negative while CF susceptibility is mostly
positive in satellite data.

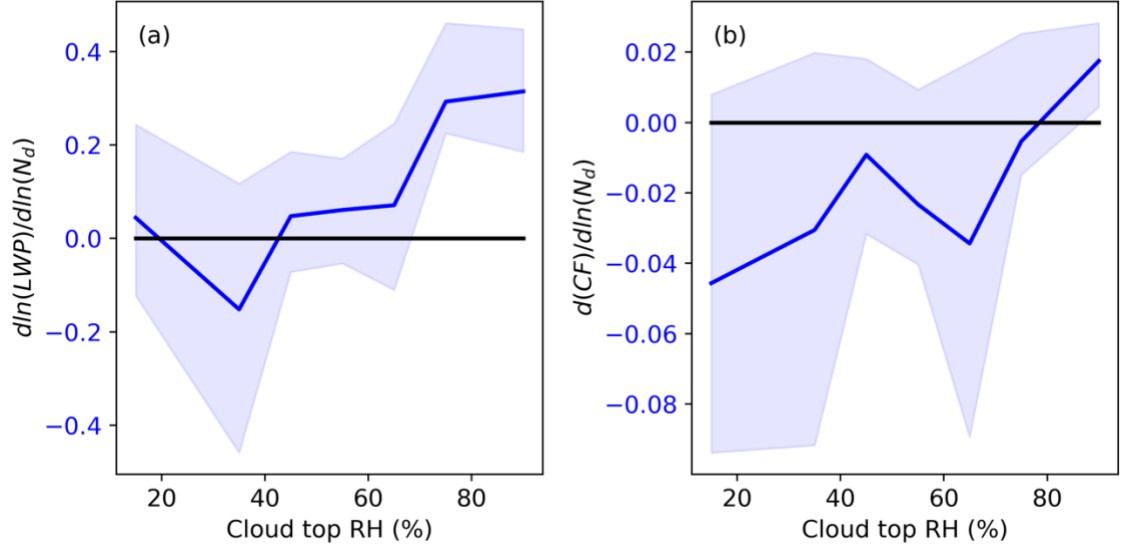

Figure 12. Dependence of (a) LWP susceptibility and (b) CF susceptibility on cloud top relative
humidity in WRF simulations during the daytime. The solid blue line shows the median value of
each RH bins and the shaded area shows the lower and upper 25[th] percentiles.
Based on the relationship between cloud susceptibility and cloud-top RH, the over-
estimated cloud-top RH of 8% may lead to an overestimation of 0.04 and 0.005 in LWP and CF
susceptibility, respectively. Meanwhile, the under-estimated cloud-top height of 480m could
result in an under-estimation of LWP and CF susceptibility of 0.18 and 0.02, respectively
(figures not shown). Future modeling studies over the ENA region need to improve the initial
and boundary conditions, e.g., through data assimilations.
To further illustrate the influence of cloud-top evaporation on LWP and CF adjustment
rate, we analyzed the relationship between cloud susceptibilities and change in the cloud-layer
buoyancy flux. As shown in the case study, buoyancy flux increases with aerosol perturbation in
precipitating clouds due to precipitation suppression, whereas it decreases in non-precipitating
clouds due to enhanced entrainment driven evaporation. Thus, changes in buoyancy flux serves
as a proxy for both cloud-top evaporation and precipitation suppression effects.

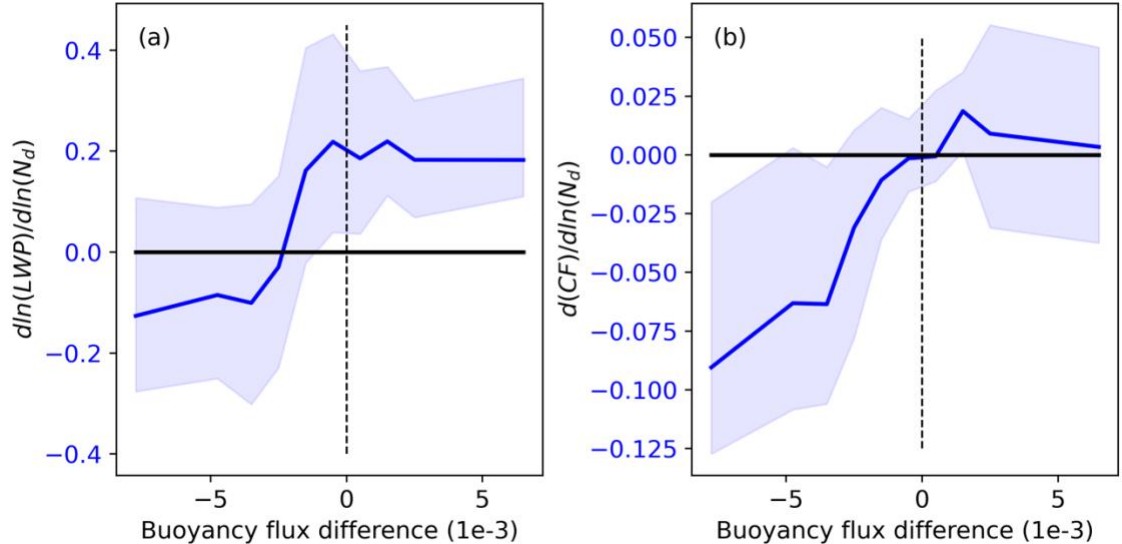

Figure 13. Dependence of (a) LWP susceptibility and (b) CF susceptibility on changes in buoyancy
flux in the cloud layer in WRF simulations during the daytime. The solid blue line shows the
median value of each buoyancy flux bins and the shaded area shows the lower and upper 25th
percentiles.
In Figure 13, changes in cloud-layer buoyancy flux is calculated as the difference in
domain-mean values between polluted and clean experiments (e.g., N=1000 vs. N=100, N=500
vs. N=100, and N=1000 vs. N=500), averaged in the cloud layer defined by the domain-mean
cloud water mixing ratio. As shown in Figure 13, two distinct regimes emerge: when cloud-layer
buoyancy flux substantially decrease with increasing aerosols, both LWP and CF decrease; when
changes in buoyancy flux is small negative or positive, LWP and CF susceptibilities are
generally positive or near zero. These results, together with those in Figure 12, support the
conclusion that the reduction in LWP and CF in the model is primarily driven by cloud-top
evaporation associated with enhanced entrainment. The absence of negative LWP responses in
earlier modeling studies may be attributed to inadequate resolution of the interactions among
boundary layer turbulence, entrainment, and cloud-top evaporation.
**3.3.3 LWP Adjustment from Internal Cloud Processes and Precipitation Heterogeneity**
In addition to model biases in representing precipitation processes and PBL thermodynamic
profiles, one leading factor contributing to the discrepancy in ACI estimates lies in how ACI is
diagnosed in numerical studies versus observations. In model simulations, ACI can be isolated
using controlled experiments by varying aerosol concentrations while holding meteorology
constant. In satellite-based analysis, however, the retrieved ACI signal inevitably includes not
only aerosol-induced cloud responses but also $N_d$–LWP covariability arising from internal cloud
processes, even under strict spatial and temporal sampling constraints. Diagnosing these internal
cloud processes in satellite observations is difficult because key governing variables, such as
cloud-base updraft speed, TKE, entrainment rate are not directly measured or retrieved. In
contrast, model simulations allow us to quantify the $N_d$–LWP relationships driven by internal
cloud processes by examining their spatial covariation under homogeneous aerosol conditions.
To ensure consistency with satellite methodology and suppress small-scale cloud heterogeneity,
pixel-level model outputs are aggregated to a 25 km × 25 km grid.

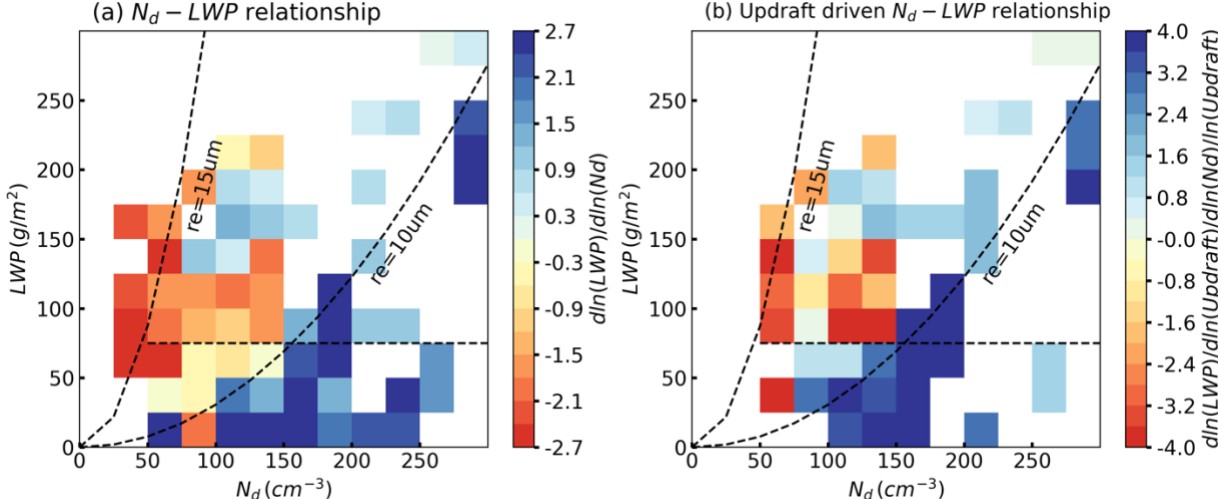

Figure 14. (a) LWP-$N_d$ relations stem from internal cloud processes (b) LWP-$N_d$ relations driven
by cloud base updraft speed in WRF simulations during the daytime.
Figure 14a shows the resulting $N_d$–LWP relationships across all cases and all aerosol
concentrations, revealing opposing signs between different cloud regimes: a strong positive
correlation for non-precipitating clouds and a strong negative correlation for precipitating clouds.
To understand this contrast, we examine whether both $N_d$ and LWP co-vary with a third
parameter indicative of internal dynamics. Cloud-base updraft speed emerges as a physical
meaningful driver: the ratio of $\frac{dln(LWP)}{dln(Updraft)}$ to $\frac{dln(N_d)}{dln(Updraft)}$ in Figure 14b closely mirrors the $N_d$–
LWP relations in Figure 14a. This indicates that cloud base updraft speed largely governs the
opposing responses. In non-precipitating clouds, stronger updrafts enhance supersaturation,
activation, and condensation, increasing both $N_d$ and LWP, and resulting in a positive $N_d$–LWP
relationship. In precipitating clouds, stronger updrafts increase LWP and rain rate, but
precipitation formation reduces $N_d$ via coalescence and collection, leading to a negative relation.
Furthermore, mesoscale variability in precipitation structure can further modulate the
$N_d$–LWP relationship in precipitating clouds. To test this hypothesis, precipitating cases
(domain-mean precipitation fraction > 0.1) are further divided into heterogeneous and
homogeneous categories based on the spatial standard deviation of precipitation fraction using
the upper and lower 50th percentile, respectively (Figure 15). Precipitation fraction is defined as
the areal fraction of cloud pixels with the column maximum reflectivity greater than –15 dBZ
(Figure 6).
In heterogeneous convective precipitation (Figure 15a), strong and spatially variable
latent heating release enhances buoyancy within clouds, while rain evaporation and downdrafts
generate cold pools. Both processes act to intensify updrafts, which in turn promote rapid droplet
growth and increase the cloud's capacity to retain liquid water, leading to higher LWP and
precipitation. Meanwhile, stronger coalescence and precipitation scavenging reduce $N_d$. Such
opposite changes in LWP and $N_d$ amplify the negative $N_d$–LWP relationship (Figure 15c). In
homogeneous stratiform precipitation, latent heating is more spatially uniform and stratification
inhibits localized buoyancy-driven updrafts. Weaker coalescence and less efficient scavenging
lead to a less negative $N_d$–LWP relationship (Figure 15d).

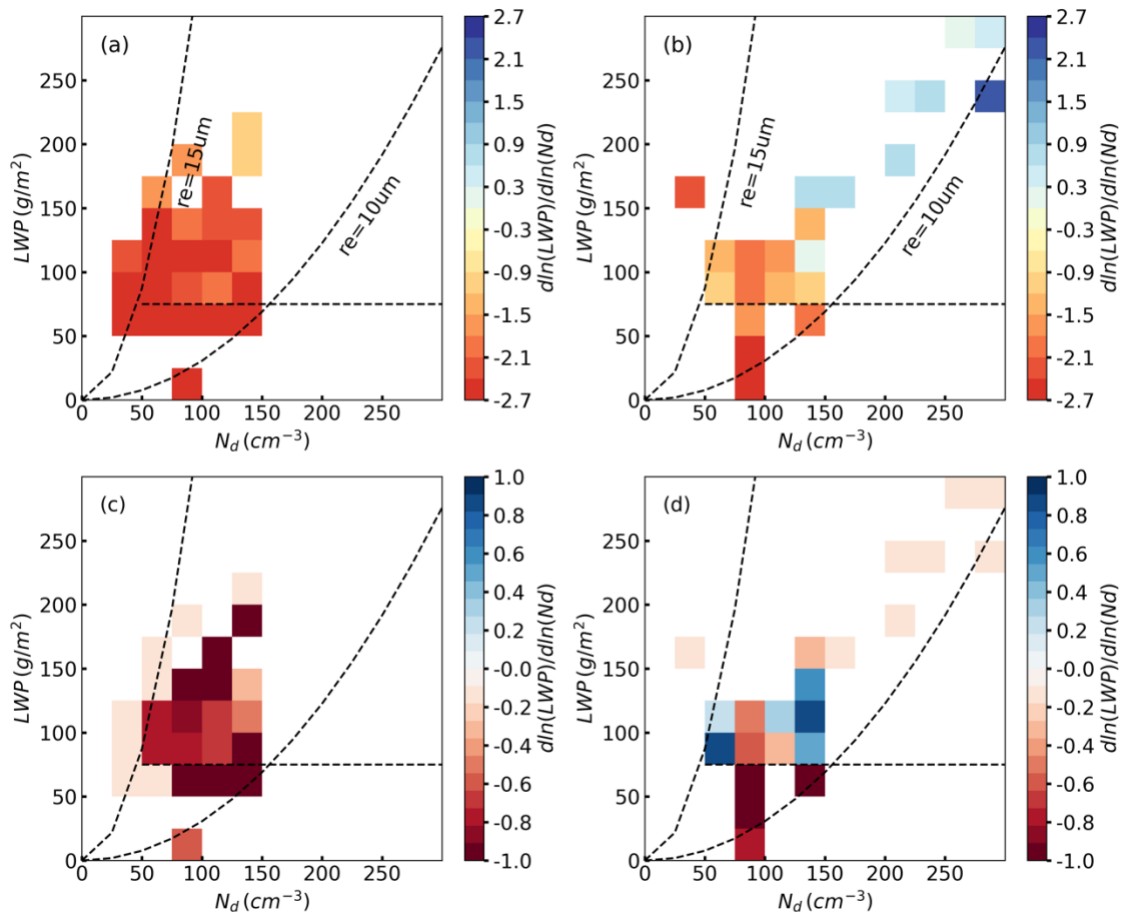

Figure 15. Same as Figure 14, but for scenes with (a) heterogeneous and (b) homogeneous
precipitation fraction. (c) and (d) show the difference between (a) and (b) with Figure 14a.
In summary, even though clouds with LWP > 75 g/m² and $r_e$ < 15 μm are typically
classified as non-precipitating thick clouds in observational ACI studies, pixel-level data real that
20–35% of these clouds produce precipitation (Figure 6a). The strongly negative LWP
susceptibilities inferred from satellite data for non-precipitating thick clouds may partly arise
from internal cloud processes driven by updraft speed and mesoscale precipitation structure,
rather than from aerosol–cloud interactions alone. providing a plausible explanation for the
model–observation discrepancy. Meanwhile, non-precipitating thin clouds with LWP < 75 g/m²
and $r_e$ < 15 μm exhibit low pixel-level precipitation fractions (typically < 0.1, Figure 6a), and the
positive $N_d$–LWP relationships arising from internal cloud processes may bias satellite-derived
LWP susceptibility toward more positive values, further expanding the model-observation gap.
The opposing signs of $N_d$–LWP relationships in Figures 14a and 5c for non-precipitating thin
clouds highlight the need for additional process-level analysis in future study.
**4. Conclusions and Discussions**
Previous studies found that model simulations and observations often reveal opposing
results in LWP responses to aerosol perturbations for MBL clouds. For example, satellite-based
assessments indicate a decrease of cloud LWP with aerosol perturbations, especially in polluted
conditions for non-precipitating clouds (e.g., Gryspeerdt et al. 2019; Toll et al., 2019; Zhang et
al., 2022, 2023; Qiu et al., 2024; Yuan et al., 2023; 2025). On the other hand, most GCMs and
CPMs simulate an increase of LWP with increasing aerosols (e.g., Ghan et al., 2016; Michibata
et al., 2016; Mülmenstädt et al., 2024; Fons et al., 2024; Christensen et al, 2024). Previous
studies found that increasing model resolution to sub-kilometer can improve the representation of
precipitation process and model performance in ACI by resolving the small-scale process most
relevant to ACI (e.g., Terai et al., 2020). It remains unclear how well models perform at close to
LES scale in representing the ACI feedback when using realistic meteorological conditions and
large case ensembles across various cloud states and synoptic regimes.
To address these gaps, our study makes three key advances: (1) we conduct a series of
realistic near-LES-scale case studies that enable direct comparison with ground-based and
satellite observations to reconcile observed–modeled discrepancies; (2) we examine a large
ensemble of MBL cloud cases spanning a range of cloud states and synoptic conditions to
capture the diversity of ACI responses; and (3) we use the same two-moment microphysics
scheme implemented in several GCMs and CPMs, making our findings directly relevant for
improving microphysical parameterizations in climate models.
The simulated MBL clouds generally match the satellite observation in domain mean
cloud coverage and mesoscale organization (Figures 1, 3, S2-S4), while the model may struggle
to capture the diurnal evolution of clouds, especially the dissipation of clouds in the afternoon.
Model overestimate cloud LWP, especially in the polluted runs and underestimated cloud top
height compared to satellite retrievals. To show the dependence of cloud responses on cloud
state, LWP susceptibilities are displayed in the $N_d$-LWP parameter space (Figure 5). For non-
precipitating thin clouds, our simulations show a consistently negative but weaker LWP
susceptibility compared to satellite observations, with a mean of−0.13. The negative LWP
susceptibility likely result from the better resolved turbulence, condensation/evaporation
processes and their feedback on PBL thermodynamics. More specifically, increases in aerosols
enhance turbulence and TKE in the cloud layer. With the dry air above, the entrained dry air
intensifies evaporation, reduces buoyancy flux in the cloud layer and leads to dissipations of
clouds (Figure 4, 13).
For precipitating clouds, our model predicts a slight increase in LWP with the mean
susceptibility of +0.15, which is consistent with the precipitation suppression hypothesis and the
climatological mean cloud response for heavily precipitating clouds (e.g., Qiu et al., 2024). For
non-precipitating thick clouds, model simulations and satellite observations show the largest
disagreement with opposite LWP susceptibilities of +0.32 vs. −0.69, respectively. Meanwhile,
the non-precipitating thick clouds are the dominant cloud state in the model, with a total
frequency of 49%, compared to a 15.7% frequency of occurrence in satellite observations. The
overestimation of $N_d$ arise from the overestimated aerosol concentration in the configuration,
combined with the absence of precipitation scavenging in the model. The overestimation of LWP
is due to the positive LWP susceptibility in thick clouds where LWP in N=100 simulation show
good agreement with satellite retrievals (Figure S9)
Our analyses indicate that such discrepancy could mainly result from the overestimation
of precipitation for thick clouds: where MBL clouds in simulations produce precipitation at
much smaller cloud drop size (e.g., $r_e > 10 \ \mu m$) and in more polluted conditions compared to
satellite observations (Figure 6). Based on ARM radar observations, our simulations reasonably
capture the non-precipitating regime and the transition from non-precipitating to drizzling clouds

within the same $r_e$ and $\tau$ range as observed (Figures 7, 8). Our simulation result appears to better represent marine clouds than GCM or GCPM, which often initiate drizzle or rain at cloud top and rarely simulate non-precipitating clouds (e.g., Jing et al. 2017, 2019; Michibata and Suzuki, 2020). However, several biases remain. In non-precipitating clouds, the model shows near-constant $Z_e$ profile with height, whereas observations show a decrease near cloud top, suggesting an underestimation of entrainment and evaporation (Figure 8). In thicker clouds ($\tau > 20$), drizzle often initiates too early at cloud top ($Z_e > -15$ dBZ), indicating excessive autoconversion. This early onset allows raindrops to grow too large through prolonged collection in deeper clouds, resulting in overestimated rain rates ($Z_e > 0$ dBZ), whereas observations show only drizzle (Figure 8). Additionally, stronger rain in polluted cases with large $r_e$ points to an overly broad DSD, as the dispersion parameter η in the Morrison scheme increases with $N_d$, and the DSD slope flattens with larger $r_e$ (Figures 9, 10). The overestimation of precipitation for thick clouds results in an increase in LWP from precipitation suppression in the simulation.

The overestimation of LWP susceptibility may also stem from biases in ERA5 and WRF profiles in representing the location and strength of moisture inversions (Figures S6, S8), leading to shallower PBL and a moist bias above the clouds in the simulations (Figure 11). Consistent with observations, model simulations show a positive correlation between LWP susceptibility and cloud-top RH, suggesting that the wet bias in cloud-top RH contributes to the positive bias in LWP susceptibility (Figure 12).

Lastly, we find that part of the discrepancy in quantified ACI may stem from $N_d$-LWP relationships driven by internal cloud processes that are mixed with the ACI signals in satellite observations. Using model simulations with homogenous aerosol concentrations, we isolate these internally driven $N_d$-LWP relationships. Our results reveal large opposing signals between precipitating clouds (large negative relationships) and non-precipitating clouds (large positive relationships), primarily governed by cloud base updraft speed (Figures 14) and modulated by mesoscale cloud and precipitation organization (Figure 15). Therefore, the strongly negative LWP susceptibility observed in thick clouds in satellite data could reflect internal cloud dynamics rather than true ACI.

This study shows that while the discrepancy in ACI assessments between observations and models can be reduced by increasing model resolution for precipitating and non-precipitation thin clouds, the positive bias in the LWP susceptibility for non-precipitating thick clouds persists. This bias is attributed to parameterization deficiencies in the microphysics scheme and model biases in lower tropospheric thermodynamics over the ENA region. These findings may motivate improvements in precipitation parameterizations and encourage their process-level evaluation against observations.

**Data availability:**

The WRF model used, version 4.2.2, is freely available from the developers' website (https://github.com/wrf-model/WRF/releases, WRF, 2022). SEVIRI Meteosat cloud retrieval products, produced by NASA LaRC SatCORPS group, are available from the Atmospheric Radiation Measurement (ARM) Data Discovery website at https://adc.arm.gov/discovery/, Minnis Cloud Products Using Visst Algorithm. The ARM ground-based radar and lidar observations (KAZRARSCL), LWP retrievals, and balloon sounding observations are available from ARM Data Discovery.


**Acknowledgments:**


We are grateful to the Atmospheric Radiation Measurement (ARM) user facility, a U.S. Department of Energy
(DOE) Office of Science user facility managed by the Biological and Environmental Research Program for
providing ARM observation data and archiving SEVIRI Meteosat cloud retrieval products. We mainly used the
computing resources from the National Energy Research Scientific Computing Center (NERSC), which is supported
by the Office of Science of the U.S. Department of Energy under Contract No. DE-AC02-05CH11231. This work
was performed under the auspices of the U.S. DOE by LLNL under contract DE-AC52-07NA27344. LLNL-JRNL-
2008226. PNNL is operated by Battelle for the U.S. Department of Energy under Contract DE-AC05-76RLO1830.
Xiaoli Zhou acknowledges the funding support from the Canada First Research Excellence Fund Transforming
Climate Action program (TCA-LRP-20241-1.1-05).

**Author contribution:**


SQ carried out the data analysis and wrote the manuscript. XZ and HL ran the simulations. PW provided the ground-
based cloud microphysics retrievals. All authors contributed to the design of the study, interpretation of the results,
and edit the manuscript.

**Financial support:**


This work is supported by the DOE Office of Science Early Career Research Program and the ASR Program.

**Competing interests:**


The authors declare that they have no conflict of interest.

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
