# Peer review of "Understanding the Causes of Satellite–Model Discrepancies in"

_EGUsphere, 2025_

## Referee Comment (RC2)

Review of "**Reconciling Satellite-Model Discrepancies in Aerosol-Cloud Interactions Using Near-LES Simulations of Marine Boundary Layer Clouds**" by Shaoyue Qiu, Xue Zheng, Peng Wu, Hsiang-He Lee, and Xiaoli Zhou

This manuscript identified model bias on LWP responses to aerosol perturbations and potential causes behind such bias using near-LES simulations and multiple observations over the Eastern North Atlantic region. By comparing the modelled LWP susceptibility with satellite observations, they found that modelled LWP susceptibility from non-precipitating, thick clouds have the largest discrepancy compared to the observations, while the LWP susceptibilities from precipitating and non-precipitating thin clouds show relatively good agreements with observations. It is suggested that the model overestimates precipitation for thick clouds including excessive autoconversion and accretion, and underestimates entrainment and evaporation, which are the main reasons for the LWP susceptibility discrepancy in these non-precipitating thick clouds. They also found that the modelled cloud susceptibilities are sensitive to cloud top humidity, and the bias of cloud top humidity in the model can be another reason for the LWP susceptibility discrepancy.

The findings in this manuscript are insightful and important for improving representation of aerosol-cloud interactions in the models. The topic and research questions are also relevant within the scope of ACP. However, I have several major comments outlined below for the improvement of this manuscript, and I recommend resubmission after the following comments are addressed.

Recommendation: major revisions

Major comments:

- I am concerned about the ability of the model to simulate LWP for the selected cases. In Figure 5, the model simulates non-precipitating, thick clouds with high LWP much more frequently than the Meteosat observed. These non-precipitating, thick clouds are key to the later-on analysis and conclusions. Comparison of LWP between model and observation is only for two cases, and Figure S2-S4 only provide a qualitative comparison of cloud fields. I suggest a more quantitative model-observation comparison for the selected cases, and a more detailed description and explanation on the LWP bias (currently there is only one sentence at Line 422 stating the potential reason of lack precipitating scavenging feedback on aerosol and $N_d$) and how this bias affects your conclusions. Although constant aerosol number concentrations are used for simulations, it will be helpful to have the $N_d$ comparison as well.

- The "Data and methodology" section needs more details. For observational data, what are the specific products or variables used from satellite? What are the uncertainties of your observations and how good are they? How did you calculate $N_d$ from Meteosat, what is the assumptions and uncertainties of the selected method on your cases?

For WRF model, how are the key warm cloud processes treated in your model, what are the parameterizations and what are the limitations of these treatments for your cases? What is the limitation of using a constant total aerosol number concentration throughout the domain for your model-observation comparison on LWP susceptibility? What is the default value you selected for aerosol number concentrations for your cases and are they the same for all cases? How did you quantify the $N_d$-LWP relationships driven by internal cloud processes and by cloud base updraft speed?

- Naming of model simulations are unclear and sometimes confusing throughout the manuscript. Currently they are described with "polluted" and "clean" in comparison. This can be misleading when you switch to another set (e.g., N=500 can be "clean" compared to N=1000 but can be "polluted" compared to N=100). In addition, "clean" is also used for describing observations (Line 497) and there is also a description of "ultra-clean" (Line 560) for the N=100 simulation. I suggest a consistent name for each model configuration in the manuscript for clarity.

- How do different synoptic regimes affect the LWP susceptibility? You mentioned to investigate the variation of ACI across different synoptic conditions in the Introduction (Lines 117-120) and therefore chose these 11 cases, however little results and analysis are shown in this manuscript on this question.

- Many captions in this manuscript are not complete and refer to captions in another figure. I suggest to include full captions for all the figures and be clear about the data used in the figure.

- In Section 3.3.1 Precipitation Efficiency, there are many comparisons between model and ground-based observations for cloud with different $R_e$ and optical depth. However, the current Figures 7-10 are for observations, N=100, N=500, N=1000 and each has 9 subplots categorized by $R_e$ and optical depth, making the whole section sometimes hard to follow. It might be helpful to reorganize these figures and perhaps paragraphs as well, so that observation and all model results are in the same figure for comparison. For example, Figure 7 can just contain clouds with optical depth less than 10 and the column now becomes observation, N=100, N=500, and N=1000. Or separate the figures by non-precipitating, drizzle and rain.

Minor comments:

- Line 1: I don't think "reconciling" is accurate for the title of this manuscript. I think key processes and reasons behind the inconsistent LWP susceptibility are identified in this manuscript, but this issue is not resolved here and requires model improvement.
- Lines 18-19: "largely due to" – I don't think incorrect LWP responses to aerosol perturbations is the reason but a main issue. The reasons can be poor representation of aerosol and cloud processes.
- Line 25: "a modest LWP decrease" to an increase in $N_d$.

- Line 26: "In contrast" to? It feels coming from nowhere. If you would like to suggest that non-precipitating thin clouds have consistent LWP susceptibilities from model and observation, but not for non-precipitating thick clouds, then you need to state this clearly.
- Line 108: please define the abbreviation of "MBL".
- Lines 128-133: What are the specific cloud retrievals and what are the uncertainties of each cloud retrieval? In addition, you have the method of calculating $N_d$ from satellite mentioned at Line 386-392, but I think it will be better to move to this section. It is also useful to include version numbers of satellite product here and in the Data availability section.
- Line 138: How was the satellite retrieval smoothed to 25-km resolution?
- Line 155: "0000 UTC"
- Lines 163-169: ERA5 data is not observational data but reanalysis data, therefore I don't think this should be described here under the observational data subsection. It can be put in a separate subsection, or you can change the name of this subsection to something like "Datasets" and separate into satellite data, ground-based data and reanalysis data.
- Lines 180-182: What are the spatial resolution of the other two nested domains?
- Line 186: How often is the lateral boundary condition updated?
- Lines 189-191: How are boundary layer and clouds treated in the innermost domain?
- Figure 1: How does Meteosat retrieve cloud coverage and is the modelled cloud cover comparable to the Meteosat-retrieved cloud coverage? How is cloud top height defined in model output and how does Meteosat retrieve cloud top height? I suggest adding time series of $N_d$ here. In addition, how does N=500 simulation look like?
- Lines 291-292: I don't think the cloud coverage from N=100 simulation closely matches the observed cloud coverage, but underestimates the cloud cover. It will be helpful to add some numbers here as well, rather than just quantitative descriptions.
- Lines 292-294: Can you suggest the reasons behind the model failed to simulate the dissipation of clouds? And how may this bias affect the modelled LWP susceptibility?
- Figure 3: Please use a full caption here rather than referring to another figure's caption. Similar to the comments for Figure 1, I suggest adding time series of $N_d$ here as well.
- Figure 4: Please use a full caption here.
- Figure 5: "WRF simulations" are these from all polluted versus clean simulations or just one of the sets? Are $R_e$ on these plots from the model or from satellite? Please make sure the axes are same for the model and observation plots. Currently they are different and make it difficult to compare with.
- Line 382: How does the Meteosat LWP susceptibility calculated?
- Line 386: What does it mean by "to be consistent with satellite observations"?
- Line 395: I think it will be useful to add a sentence here on how you define different types of clouds: precipitating versus non-precipitating, thick versus thin.
- Lines 407-410: Your satellite observations for precipitating clouds are different from your simulations and previous study with long-term data. Can you suggest why? Is this because of the limitations of satellite data? Does this affect your model-satellite comparison for other clouds?

- Line 429-430: I don't think Figure 5 show that the model results agree with Meteosat observations for an increase in LWP in precipitating clouds (Meteosat suggest a decrease).
- Line 434-435: If the modelled LWP response is showing large discrepancy compared to observations, this is not indicating the robustness of the results. Please explain in detail on the reasons why you suggest that the model results are robust.
- Figure 6: It is confusing here that the $r_e$ dashed lines across different $r_e$ contour colours in (a) and (c). Please be clear about how each effective radius is calculated or derived in (a), (c) and the dashed line.
- Lines 493-495: Frequencies from satellite data only sum to 90.6%, what and where are the rest 9.4%? In addition, can you explain more on why the selected cases are representative just based on the frequencies?
- Line 497 and others: what does "clean condition" mean here? You use "clean" to describe both simulations and $R_e$ condition in your figures in this section, which is confusing during reading.
- Lines 506-507: "likely due to mixing and evaporation" – can you be more specific on this?
- Figures 8-10: Please use full captions for these figures.
- Lines 558-559: I can see that DSD is compared by using percentages of $R_e$ categorizes, but it may be helpful and clearer to compare full DSD from different model simulations and observations for clouds with different optical depths.
- Lines 591-599: The description of DSD in the model is better to be put in the Data and Methodology section along with the descriptions of other treatments of warm cloud processes.
- Lines 636-638: The cloud tops are defined differently in ARM observations and in the model. Since you have the model radar simulator, why not using the same definition here based on the radar reflectivity profile for observed and modelled cloud tops?
- Lines 644-648: I commend the authors on considering the spatial representation issue and it will be helpful to describe how the temporal representation issue is treated, e.g., what are the model output time for comparing cloud top RH with the sounding observations?
- Figure 12: What is the shaded area for?
- Line 656: "in the simulations"- are these for all simulations with all aerosol number concentration or specific ones? Does the dependence of these cloud susceptibilities on cloud top relative humidity change when using different sets of simulations (e.g., between N=1000 vs. N=100 and N=1000 vs. N=500)?
- Figure 13: It will be helpful to have vertical lines where buoyancy flux difference equals to 0 as well on the plot. Similar to the comment on Figure 12, what is the shaded area for?
- Line 691: please explain the $N_d$-LWP here in detail.
- Figure 14: Are these from simulations with all different aerosol concentrations?
- Figure 15: Please use a full caption.
- Lines 745-754: I think several references are missing here in this first paragraph when mentioning the findings from previous studies.
- Lines 762-763: I suggest adding the LWP bias here rather than using "generally match".

- Lines 778: Are there any other potential reasons for the LWP bias and what's the reason that you suggest the lack of precipitation scavenging feedback on aerosols is likely the cause here?

---

## Author Comment (AC1)

**Responses to reviewer comments**

In this response letter, we provide a point-by-point response to each comment. The original comments are in *blue italic font*, and our response are in black font. Changes made in the manuscript are listed in quotes with line numbers of the tracked changes version.

*Reviewer #1:*

***General comments:***

*The authors employ high-resolution near-LES WRF simulations to investigate aerosol-cloud interactions (ACI) in marine boundary layer (MBL) clouds over the Eastern North Atlantic (ENA). The study is methodologically rigorous, leveraging satellite retrievals, ground-based ARM observations, and process-level diagnostics (e.g., CFODD analysis). The conclusions highlight persistent model-observation discrepancies in LWP susceptibility, particularly for non-precipitating thick clouds, and propose mechanistic explanations tied to precipitation efficiency and entrainment biases. The paper is well-organized and addresses a critical gap in ACI understanding. However, several scientific and methodological issues require clarification to ensure robustness.*

***Major comments:***

*- Section 3.1 evaluates two representative cases by Meteosat; however, the Meteosat data shows dissipation and reformation processes that the model didn't capture. I expected the authors to focus on the discrepancy and discuss the reason for this mismatch, but I couldn't find any direct discussion on this. In some paragraphs, the authors point out model bias in LWP susceptibility to aerosols, which is good, but still, the model bias in Section 3.1 needs explanations.*

Thanks for your question. The lack of cloud dissipation and diurnal variation in marine boundary layer (MBL) clouds in the WRF model is likely associated with the biases in the thermodynamic profiles inherited from ERA5. As seen in Figures R1 left figure, on 21 July 2016, ARM sounding observations indicate a sharp decrease in moisture above the PBL between 14 and 20 UTC, leading to the dissipation of clouds after 14 UTC (Figure 1a). In contrast, both ERA5 and WRF simulation show a gradual decrease in specific humidity and relative humidity above the PBL from 0 to 20 UTC, resulting in a much moister layer above clouds in the model (Figures R1 middle and right). Consequently, clouds did not dissipate in the afternoon in the simulation.

On 25 July 2016, the ARM sounding observations similarly exhibit a pronounced decrease in specific humidity and relative humidity above the PBL between 14 and 24 UTC (Figure R2). In this case, the WRF simulation accurately capture the observed feature, reproducing a sharp decrease in moisture above the PBL from 14 to 24 UTC. As a result, clouds in the N=100 and N=1000 simulations dissipate from 14 to 24 UTC, consistent with satellite observation (Figure R3). This pattern also holds for other cases in the high-ridge regime, such as 22 and 28 July 2016, where the accuracy of the simulated PBL moisture variation determined whether the model captured the observed diurnal evolution of clouds (figures not shown). These cases demonstrate that the diurnal cycle of cloudiness is highly sensitive to the representation of diurnal variation in moisture as well as the moisture gradients near the inversion.

The fixed, vertically uniform aerosol concentration further contributes to the persistence of clouds by maintaining unrealistically high CCN concentrations throughout the day and suppressing precipitation. The lack of precipitation scavenging also reduces evaporative cooling and weakens cloud–PBL decoupling, inhibiting afternoon cloud breakup.

We added these discussions to the second last paragraph in Section 3.1:

"The absence of afternoon cloud dissipation in WRF simulations are likely associated with model biases in the thermodynamic structure inherited from ERA5. For example, on 21 July 2016, ARM sounding observations show a pronounced decrease in specific humidity and relative humidity above the PBL between 14 and 20 UTC (figures not shown). This sharp drying leads to cloud erosion in the observations. However, WRF simulations or ERA5 reanalysis produces only a gradual reduction in moisture from 00 to 20 UTC (Figure 2a), maintaining a moist layer above cloud top and prevent cloud breakup.  On 22 July 2016, the model reproduces the moisture gradient above PBL with a warm and dry layer above, the lifted cloud top in the N=1000 simulation entrain dry air into cloud system and dissipate clouds in the afternoon (Figure 3a). On days when ERA5 accurately capture the observed moisture decrease above PBL (e.g., 25 and 28 July 2016), the model reproduces both the dissipation and evening redevelopment of clouds seen in Meteosat data (figures not shown). This indicates that the diurnal evolution of MBL clouds is highly sensitive to the representation of diurnal variation in moisture as well as the moisture gradients near the inversion.

The prescribed, vertically uniform aerosol concentration further reinforces cloud persistence by maintaining elevated CCN levels and suppressing drizzle formation. The lack of precipitation scavenging prevents cloud-base evaporative cooling and inhibits decoupling, both of which would otherwise promote afternoon cloud breakup. The implications of thermodynamic and aerosol-related biases for the estimated ACI are discussed in detail in Section 3.3.2. (Lines 436-455)"

[Figure]

Figure R1 Time series of thermodynamic profiles on 21 July 2016, for (a) potential temperature (unit: K) (b) specific humidity (unit g/kg), (c) relative humidity in (left) ARM interpolated sounding, (middle) ERA5 reanalysis, and (right) in WRF N=100 simulation.

[Figure]

Figure R2 Same as Figure R1 but for the case on 25 July 2016.

[Figure]

Figure R3. Time series of domain-averaged cloud properties from observations and model simulation on 25 July 2016. (a) Cloud coverage, (b) cloud top height, (c) cloud liquid water path, and (d) rain-water path for N=100 (blue lines) and N=1000 (orange lines) experiments.

*- Paragraph in Lines 395: the authors use 15 microns as the threshold to differentiate precipitating clouds and non-precipitating clouds. Since the authors have very good representative cases with one precipitating and another not, why not separate the two scenarios by cases? I believe the authors know the threshold is a bit tricky because other values (12 microns or 13 microns) have been used in previous literature, and so far, we don't have an agreement on which number is best.*

Thanks for the suggestion. We didn't separate the non-precipitating and precipitating scenarios by cases because most cases have clouds transitioned from one to another during the simulation period. Instead, we classify cloud state at each time step. We use the 15-micron threshold in the model to be consistent with the precipitation threshold used in the satellite observations. As the main goal of this study is to explain the discrepancy between the observed and simulated LWP susceptibility, we use the same classification of precipitation in the model as in the satellite observation to make consistent comparison. In Figure 6d, we evaluated the 15-micron threshold in the model using the column maximum radar reflectivity ($Z\ max$) greater than $-15$ dBZ at each model output time. As shown in Figure 6d, model generates precipitation too often at smaller drop size with $r_e > 10\ \mu m$ and at higher $N_d$ concentration. The over-estimation of precipitation in the model is the leading cause of the positive bias in LWP susceptibility.

We agree that the threshold of 15-micron could be tricky and different threshold values have been used in previous studies. In our previous satellite observational study, we

evaluated different effective radius thresholds and rain rate thresholds in satellite retrievals using precipitation masks derived from ground-based radar reflectivity at the ENA site, and we found that the $r_e > 15\ \mu m$ threshold showed the best agreement with ground-based observations (Qiu et al., 2024). To address this comment, and a similar comment from the other reviewer, we add the definitions of different cloud states:

"Based on the relationships between $r_e$, LWP, and $N_d$ in the satellite retrievals (e.g., $LWP = \frac{4r_e\tau}{3Q_{ext}}$, $N_d = \frac{\sqrt{5}}{2\pi k}(\frac{f_{ad}c_w\tau}{Q_{ext}\rho_w r_e^5})^{1/2}$), $r_e = 15$ isolines is marked in the LWP-$N_d$ parameter space as an commonly used indicator of precipitation likelihood in the satellite retrieval (e.g., Gryspeerdt et al., 2019; Toll et al., 2019; Zhang et al., 2022; Qiu et al., 2024). Based on the distinct LWP, cloud albedo and CF susceptibilities, MBL clouds are classified into three states: the precipitating clouds ($r_e > 15\ \mu m$), the non-precipitating thick clouds ($r_e < 15\ \mu m$, LWP$> 75\ gm^{-2}$), and the non-precipitating thin clouds ($r_e < 15\ \mu m$, LWP$< 75\ gm^{-2}$) (Qiu et al., 2024). To be consistent with observational reference, the WRF simulated cloud states are classified using the same definition. (Lines 479-487)"

*- LWP Susceptibility Discrepancy (Lines 417–419)*
*The model shows a positive LWP response (+0.32) for non-precipitating thick clouds, while observations show a strong negative response (-0.69). What is the primary driver of this discrepancy?*
Thanks for your question. We added more explanation in the summary paragraph of section 3.2 to address this comment:

"Large discrepancies remain for non-precipitating or lightly drizzling thick clouds, where the model simulates too many polluted thick clouds and yields an opposite (positive) LWP response compared to the strongly negative satellite signal.

In addition, the model-observation discrepancy persists across all synoptic regimes, suggesting that they originate from the model's representation of cloud microphysics, precipitation, and aerosol-cloud coupling rather than from large-scale meteorological variability. The robustness of these modeled LWP response, consistent with previous LES studies of similar cloud regimes (e.g., Wang et al., 2020; Lee et al., 2025), further motives the central focus of the next section: diagnosing the physical mechanisms driving these biases. We show that three leading factors dominate the discrepancy: excessive precipitation production in thick clouds, a moist bias above cloud top, and satellite retrieved $N_d$-LWP relationships contaminated by internal cloud processes. (Lines 592-608)"

*- Model Biases and Initial Conditions (Lines 356-360, 642-664)*
*The study identifies biases in ERA5 reanalysis (e.g., underestimated PBL height, overestimated cloud-top RH) as a significant source of discrepancy. However, the extent to which these biases propagate into the WRF simulations and affect ACI estimates is not fully quantified. Sensitivity tests using alternative reanalysis datasets or perturbed initial conditions could help isolate the impact of these biases. If perturbed simulations or using different reanalysis datasets add too much work, at least discussion on this point is necessary.*
Thanks for the insightful question and suggestion. As seen in Figure 11, the cloud-top RH in WRF simulations is ~ 8% higher than ARM observation. Based on the relationships between cloud-top RH against LWP and CF susceptibilities, the 8% wet bias may lead to an overestimation of 0.04 and 0.005 in LWP and CF susceptibility, respectively (Figure R4). Similarly, LWP and CF susceptibilities positively correlate with cloud top height (Figure R5). As seen in Figure R6, due to the under-estimation of PBL height in ERA5 reanalysis and WRF simulations, the simulated cloud top is ~480 m lower than Meteosat retrievals. This under-estimation may lead to an ender-estimation of LWP and CF susceptibility of 0.18 and 0.02, respectively. To conclude,

the under-estimation of cloud-top height in WRF simulations may exhibit larger impact on LWP susceptibility than the overestimation of clout-top RH, due to the larger bias in cloud-top height between simulations and observations. We have added the following discussions to the manuscript.

"Based on the relationship between cloud susceptibility and cloud-top RH, the over-estimated cloud-top RH may lead to an overestimation of 0.04 and 0.005 in LWP and CF susceptibility, respectively. Meanwhile, the under-estimated cloud-top height of 480 m could result in an under-estimation of LWP and CF susceptibility of 0.18 and 0.02, respectively (figures not shown). Future modeling studies over the ENA region need to improve the initial and boundary conditions, e.g., through data assimilations. (Lines 890-895)"

[Figure]

Figure R4. Dependence of (a) LWP susceptibility (b) CF susceptibility on cloud-top relative humidity in WRF simulations during the daytime. The solid blue line shows the median value of each RH bins and the shaded area shows the lower and upper 25$^{th}$ percentiles.

[Figure]

Figure R5. Dependence of (a) LWP susceptibility (b) CF susceptibility on cloud-top height in WRF simulations during the daytime. The solid blue line shows the median value of each RH bins and the shaded area shows the lower and upper 25$^{th}$ percentiles.

[Figure]

Figure R6. Dependence of (a) LWP susceptibility and (b) CF susceptibility on cloud top height in WRF simulations during the daytime. The solid blue line shows the median value of each RH bins and the shaded area shows the lower and upper 25th percentiles.

*- Precipitation Parameterization (Lines 540-560, 621-628)*
*The overestimation of precipitation in thick clouds is attributed to autoconversion, accretion, and DSD issues. While the analysis is thorough, the study could benefit from testing alternative microphysics schemes (e.g., P3, Thompson) to assess the robustness of the conclusions. At lease, I suggest to add discussions on the choice of microphysical schemes and parameters.*

Thank you for the insightful question. We added a paragraph on discussion of microphysics scheme at the end of Section 3.3.1:

“While our analysis focuses on the two-moment Morrison scheme, Christensen et al. (2024) found that the choice of microphysics and PBL schemes accounts for only about 30 % of the variability in simulated ACI, much smaller than the variability across meteorological conditions and cloud states. Since this study encompasses 11 cases spanning diverse synoptic regimes and cloud types, the overall conclusions are unlikely to change substantially with alternative two-moment bulk microphysics schemes. Nonetheless, future investigations using multiple microphysics schemes would be valuable for quantifying the robustness of the precipitation parameterization and its role in ACI uncertainty. (Lines 837-844)”

*- Internal Cloud Processes vs. ACI (Lines 684-737)*
*The discussion on internal cloud processes (e.g., updraft-driven -LWP relationships) is insightful but could be strengthened by explicitly separating these effects from true ACI in the observational analysis. For example, using conditional sampling (e.g., stratifying by updraft strength) might help disentangle these contributions.*

Thanks for the suggestion. In satellite observations, it is a bit challenging to disentangle the $N_d$-LWP relationships contributed by internal cloud processes from the true ACI, and we don’t have direct measurements or retrievals of the cloud base updraft speed or other direct measurement indicating internal cloud processes. To compensate for this limitation in observations, we used model outputs to quantify the $N_d$-LWP relationships contributed by internal cloud processes in section 3.3.3. We added a sentence to clarify this: “Diagnosing these internal cloud processes in satellite observations is difficult because key governing variables, such as cloud-base updraft speed, TKE, entrainment rate are not directly measured or retrieved. In contrast, model simulations allow us to quantify the $N_d$–LWP relationships driven by internal cloud processes by examining

their spatial covariation under homogeneous aerosol conditions at each timestep. (Lines 930-933)"

*The 11 cases span different synoptic regimes, but the rationale for selecting these specific cases (e.g., why not include southerly wind conditions?) is not fully explained. A further discussion on the implications of these synoptic differences would be beneficial.*

[Figure]

Figure R7. Mean liquid water path (LWP) susceptibility from WRF simulations for (a) (b) the high-ridge regime and (c) (d) the post-trough regime. (a) (c) cloud LWP susceptibility $dln(LWP)/dln(N_d)$, (b) (d) frequency of occurrence of sample in each bin.

Thanks for the question. Previous studies using ARM observations at the ENA site found that aerosol, CCN, cloud properties, and PBL properties are influenced by local emission from the Graciosa Island during southerly wind conditions (e.g., Ghate et al, 2021, 2023). As our study used radar reflectivity profiles at the ARM ENA site to evaluate simulated precipitation processes, we focus on times when the site is dominant by northerly wind from the ocean to minimize influence from the island.

Thanks for the suggestion on adding a discussion on the influence of different synoptic regimes on LWP susceptibility. As we only have one case in the "weak-trough" regime (Table S1), we compared the LWP susceptibility and the occurrence frequency of different cloud states between the "high-ridge" and "post-trough" regimes, as shown in Figures R11.

In our previous study using six-year ground-based observations at the ARM ENA site, Zheng et al. (2025) found that the "high-ridge" regime has significantly more single-layer stratocumulus clouds, thinner cloud depth, smaller LWP, and smaller surface rain rate compared to the "post-trough" regime. Consistent with our previous study, there are more non-precipitating thin clouds in the high-ridge regime compared to the post-trough regime, with the total frequency of occurrence of 49% and 40%, respectively (Figures R11b and d). For cloud susceptibility, the non-precipitating thin clouds in the high-ridge regime exhibit more negative LWP susceptibility compared to clouds with similar LWP and Nd in the post-trough regime, likely due to the cold dry air above clouds with the subsidence in the high-ridge regime. Additionally, the non-precipitating or slightly drizzling thick clouds in both regimes exhibit strong positive LWP susceptibilities, indicating that the model-observation discrepancy for this cloud state is consistent with different synoptic conditions and warrant further investigations in the next section.

We have added Figure R7 to the supplementary information and added the discussion on influence of synoptic regimes on LWP susceptibility to the manuscript:

"To further examine whether these discrepancies depend on large-scale meteorological conditions, we assessed LWP susceptibility across different synoptic regimes. Because only one case is available for the "weak-trough" regime (Table S1), our comparison focuses on the "high-ridge" and the "post-trough" regimes (Figure S10). The "high-ridge" regime shows a higher occurrence of non-precipitating thin clouds than the "post-trough" regime, with total frequencies of 49% and 40%, respectively (Figures S10b, d, t). This more frequent non-precipitating thin cloud in the model is consistent with our previous study based on six years of ground-based observations at the ARM ENA site, which revealed that the "high-ridge" regime favors single-layer stratocumulus clouds with shallower cloud depth and smaller LWP compared to the "post-trough" regime (Zheng et al., 2025).

In addition, non-precipitating thin clouds in the "high-ridge" regime exhibit more negative LWP susceptibilities than clouds with similar LWP and $N_d$ in the "post-trough" regime. This difference in LWP susceptibility is associated with the colder and drier air above clouds under subsidence in the "high-ridge" regime, which enhances cloud dissipation, as also demonstrated in the case study. Overall, non-precipitating or lightly drizzling thick clouds in both synoptic regimes still manifest strong positive LWP susceptibilities, suggesting that the model-observation discrepancy for this cloud state persist regardless of synoptic conditions and therefore warrants further investigation. (Lines 566-583)"

*- Radar Simulator Validation (Lines 214-220, 451-454)*
*The use of CR-SIM for radar reflectivity comparison is commendable, but the study does not explicitly validate the simulator against ARM observations for the specific cases analyzed. Including a direct comparison (e.g., scatter plots, statistical metrics) would bolster confidence in the model-observation discrepancies.*

Thank you for the valuable suggestion. Figure R8 shows the ARM radar reflectivity profiles for the 11 selected cases. As seen in Figure R8, the radar reflectivity profiles exhibit consistent characteristics as six-year data shown in Figure 7. The only difference is that clouds with $r_e = 5 - 10 \ \mu m$ and $\tau > 20$ start drizzling at cloud base for the selected cases (Figure R8i). Therefore, the difference between CR-SIM radar simulator and ARM observations can be attributed to model biases rather than to the representativeness of cases.

We added the following discussion: "To increase the sample size, we analyzed the climate-mean radar reflectivity profiles of stratocumulus and cumulus clouds observed during the summer months (June to August) from 2016 to 2021, comprising a total of 91,737 profiles. Radar reflectivity profiles derived from the selected 11 cases exhibit consistent characteristics (figure not shown). (Lines 662-666)"

[Figure]

Figure R8. Frequency of radar reflectivity as a function of in-cloud optical depth ($\tau_d$) for ARM ground-based observations during the daytime for the selected 11 cases. Different rows are for different ranges of optical depth ($\tau$): (a)-(c) clouds with $\tau < 10$, (d)-(f) clouds with $10 < \tau < 20$, (g)-(i) clouds with $\tau > 20$. Different columns are for different ranges of effective radius ($r_e$). The left, middle, and right columns are for $15 - 20\ \mu m$, $10 - 15\ \mu m$, and $5 - 10\ \mu m$, respectively. The black dashed lines in each panel denote $-15$ dBZ and $0$ dBZ, as thresholds of drizzle and rain, respectively. The percentage of sample (P) for each subgroup is denoted in the figure, with a total sample of 4648.

***Minor comments:***
*- Clarify Terminology (Lines 395-400)*
*The term "susceptibility" is used interchangeably for LWP and CF responses. Consider defining these terms more explicitly early in the manuscript (e.g., in the Abstract or Introduction).*

Thanks for the suggestion. We have added the definitions of susceptibility and method used to quantify LWP and CF susceptibilities to the method section. "In the context of ACI: cloud susceptibility quantifies how sensitive a cloud property responds to change in aerosol concentration or $N_d$. To constrain the spatial-temporal variation in meteorological conditions and cloud properties, cloud susceptibility is estimated as the regression slope between $N_d$ and cloud properties within the $1° \times 1°$ domain at each time step of satellite observations. In this study, we quantify both LWP and cloud fraction (CF) susceptibilities to $N_d$ perturbations. Because of the non-linear relations between LWP and $N_d$, the LWP susceptibility is quantified in logarithm scale as:

$dln(LWP)/dln(N_d)$ (e.g., Gryspeerdt et al. 2019; Qiu et al., 2024) and CF susceptibility is quantified as: $dCF/dln(N_d)$ (e.g., Kaufman et al. 2005; Chen et al., 2022; Qiu et al., 2024). (Lines 169-177)"

*- Equation (1) (Lines 387-394)*
*The derivation of $N_d$ from $r_e$ is not fully explained. Briefly clarify the assumptions (e.g., adiabaticity, k value) or cite a reference for the equation.*
Following your suggestion, we moved this paragraph to the method section and added the derivations and assumptions for $r_e$ and $N_d$ retrievals to the manuscript:

"In this study, we used the cloud mask, cloud effective radius ($r_e$), cloud optical depth ($\tau$), cloud liquid water path (LWP), cloud phase, and cloud top height variables in the SEVIRI Meteosat cloud retrieval product (Minnis et al., 2011, 2021). We focus on warm boundary layer clouds with cloud top below 3km and a liquid cloud phase. The $r_e$ and $\tau$ retrievals are based on the shortwave-infrared split window technique during the daytime. Cloud LWP is derived from $r_e$ and $\tau$ using the equation: $LWP = \frac{4r_e\tau}{3Q_{ext}}$, where $Q_{ext}$ represents the extinction efficiency and assumed constant of 2.0. Cloud mask algorithm is consistent with the CERES Ed-4 algorithm, as described in Trepte et al. (2019), where cloudy and clear pixels are distinguished based on the calculated TOA clear-sky radiance. Cloud top height is derived from the retrieved cloud effective and top temperature, together with the boundary-layer temperature profiles and lapse rate, as described in Sun-Mack et al. (2014). Cloud $N_d$ is retrieved based on the adiabatic assumptions for warm boundary layer clouds, based on the following equation:

$$N_d = \frac{\sqrt{5}}{2\pi k}\left(\frac{f_{ad}c_w\tau}{Q_{ext}\rho_w r_e^5}\right)^{1/2} \qquad (1)$$

In Equation (1), $k$ represents the ratio between the volume mean radius and $r_e$, and it is assumed to be constant of 0.8 for stratocumulus, $f_{ad}$ is the adiabatic fraction, $c_w$ is the condensation rate, $Q_{ext}$ is the extinction coefficient, and $\rho_w$ is the density of liquid water (Grosvenor et al., 2018). (Lines 148-165)"

*- Statistical Significance (Lines 406-417)*
*The differences in LWP susceptibility between model and observations are discussed, but statistical significance tests (e.g., t-tests, confidence intervals) are not reported. Adding these would strengthen the conclusions.*
Thanks for the helpful suggestion. Figure R9 shows the p values for the Welch t-test between Meteosat observations and WRF simulations for each bin. We further marked bins with p<0.05 with black outlines as shown in Figure R10. For most MBL clouds, the simulated LWP susceptibilities are significantly different than the satellite observations. For non-precipitating thin clouds, our simulations reproduce the decrease of LWP with weaker magnitude. Yet, the LWP susceptibilities are significantly different from satellite observations for most bins.
To address this comment, we updated Figure 5 and added related discussions in the manuscript. "For non-precipitating thin clouds, the simulated decrease in LWP with increasing aerosol concentration agrees in sign with satellite observations. However, the magnitude of this decrease is weaker, and the simulated susceptibilities remain significantly different from satellite estimates at 95% confidence level for most bins (Figure 5a, c). (Lines 529-533)"

[Figure]

Figure R9 P value for the Welch t-test between Meteosat observations and WRF simulations for each bin.

[Figure]

Figure R10. Mean liquid water path (LWP) susceptibility from (a) (b) WRF simulations and (c) (d) Meteosat cloud retrievals during the daytime. (a) (c) cloud LWP susceptibility $dln(LWP)/dln(N_d)$, (b) (d) frequency of occurrence of sample in each bin. The dashed lines indicate $r_e = 15\ \mu m$, $r_e = 10\ \mu m$, and LWP= 75 $gm^{-2}$, as $r_e$ thresholds for precipitation (precipitating clouds located to the left of the line), and for thick clouds (with LWP > 75 $gm^{-2}$), respectively. Black-outlined bins denote cases where the WRF and Meteosat LWP susceptibilities differ significantly (p < 0.05) based on a Welch's t-test.

- *CFODD Interpretation (Lines 494-523)*

*The CFODD analysis is insightful, but the physical interpretation of reflectivity slopes (e.g., why steeper slopes indicate stronger accretion) could be briefly elaborated in the text.*

Thanks for the question and suggestion. Based on the definition of droplet collection efficiency ($E_c$) for a continuous collection model, and the assumption of the relationship between radar reflectivity ($Z_e$) and cloud drop size, we can derive the relation between $Z_e$ and $E_c$ as

$$\frac{dZ_e}{Z_e} \approx \frac{\alpha}{6} E_c d\tau_d,$$

where $\alpha$ is a constant and is associated with what variable is conserved in the process, $\tau_d$ is in-cloud optical depth. For a complete derivation, please refer to Suzuki et al. (2010) study. Therefore, the slope of the reflectivity changes as a function of $\tau_d$ in the CFODD analysis contains information about the droplet collection efficiency $E_c$. To address this comment, we added the text to the manuscript:

"Based on the relationship between $Z_e$ and the droplet collection efficiency ($E_c$), the vertical slope of $Z_e$ as a function of in-cloud optical depth ($\tau_d$) is directly linked to $E_c$, a steeper slope indicates a larger $E_c$ (Suzuki et al., 2010). (Lines 648-650)"

*- Aerosol Prescription (Lines 199-200)*
*The assumption of fixed aerosol concentrations (no vertical/horizontal variability) may oversimplify real-world conditions. Acknowledge this limitation and discuss its potential impact on ACI estimates.*

Thanks for the suggestion. Yes, the uniform aerosol concentration assumption over-simplifies the spatial and temporal heterogeneity from local emission and long-range transport, the relative location between aerosol plumes and cloud, as well as processes such as wet scavenging, and the reactivation of CCN from evaporated rain drops. In a companion study, we employed the WRF model with the interactive chemistry and aerosol schemes and investigated ACI and its feedback on both clouds and aerosols using same model configuration and cases (but less) as this study (Lee et al., 2025). In Lee et al. (2025), we found a consistent positive LWP response for precipitating clouds as this study (Figure 11 in Lee et al., 2025). As we assumed a higher ratio of the Aitken mode aerosols (80% for the Aitken mode and 20% for the accumulation mode) in that study, and activated CCN and $N_d$ concentrations are much lower in Lee et al. (2025) than in this study. In addition, with the comprehensive aerosol module in WRF-Chem, we found signals of increased reactivation of CCN from evaporated raindrop due to larger aerosols in the accumulation mode.

We added the following discussion to the method section acknowledging the limitation and potential impact: "The fixed aerosol field neglects spatial and temporal variability driven by emissions, long-range transport, wet scavenging, and CCN reactivation from evaporated raindrops. These missing processes can sustain higher CCN concentrations, suppress precipitation, and potentially exaggerate positive LWP responses.

Despite this simplification, our companion WRF-Chem study (Lee et al., 2025) shows that, even with full aerosol microphysics, wet scavenging, and aerosol reactivation, the simulated LWP responses remain broadly consistent with the results presented here, especially the positive susceptibility in precipitating clouds. This agreement suggests that the key findings of this work are robust, although the prescribed-aerosol assumption may still contribute to some of the quantitative discrepancies discussed in Section 3. (Lines 250-260)"

*- Diurnal Cycle (Lines 764-765)*
*The model's struggle to capture afternoon cloud dissipation is noted but not explored. A brief discussion of potential causes (e.g., radiation biases, entrainment rates) would be helpful.*

We added the discussion on the potential causes for the model missing the diurnal variation in clouds. Please refer to major comment #1 for details.

---

## Author Comment (AC2)

**Responses to reviewer comments**

In this response letter, we provide a point-by-point response to each comment. The original comments are in *blue italic font*, and our response are in black font. Changes made in the manuscript are listed in quotes with line numbers of the tracked changes version.

*Reviwer #2:*

***General comments:***

*This manuscript identified model bias on LWP responses to aerosol perturbations and potential causes behind such bias using near-LES simulations and multiple observations over the Eastern North Atlantic region. By comparing the modelled LWP susceptibility with satellite observations, they found that modelled LWP susceptibility from non-precipitating, thick clouds have the largest discrepancy compared to the observations, while the LWP susceptibilities from precipitating and non-precipitating thin clouds show relatively good agreements with observations. It is suggested that the model overestimates precipitation for thick clouds including excessive autoconversion and accretion, and underestimates entrainment and evaporation, which are the main reasons for the LWP susceptibility discrepancy in these non-precipitating thick clouds. They also found that the modelled cloud susceptibilities are sensitive to cloud top humidity, and the bias of cloud top humidity in the model can be another reason for the LWP susceptibility discrepancy.*

*The findings in this manuscript are insightful and important for improving representation of aerosol-cloud interactions in the models. The topic and research questions are also relevant within the scope of ACP. However, I have several major comments outlined below for the improvement of this manuscript, and I recommend resubmission after the following comments are addressed.*

*Recommendation: major revisions*

***Major comments:***

*- I am concerned about the ability of the model to simulate LWP for the selected cases. In Figure 5, the model simulates non-precipitating, thick clouds with high LWP much more frequently than the Meteosat observed. These non-precipitating, thick clouds are key to the later-on analysis and conclusions. Comparison of LWP between model and observation is only for two cases, and Figure S2-S4 only provide a qualitative comparison of cloud fields. I suggest a more quantitative model-observation comparison for the selected cases, and a more detailed description and explanation on the LWP bias (currently there is only one sentence at Line 422 stating the potential reason of lack precipitating scavenging feedback on aerosol and Nd) and how this bias affects your conclusions. Although constant aerosol number concentrations are used for simulations, it will be helpful to have the Nd comparison as well.*

Thank you for the constructive comment. In Figure R1, we include a quantitative comparison between model and observed LWP and Nd. As seen in Figure R1a, the simulated LWP in the N=100 simulation agrees well with the Meteosat observation, with a mean value about 10% lower than Meteosat. However, since most of the thick clouds with LWP greater than 75 g/cm2 exhibit a positive LWP susceptibility (Figure 5a), the LWP in the N=500 and N=1000 simulations increases relative to that in the N=100 simulation.

[Figure]

Figure R1 Scatter plot of domain-averaged cloud properties from Meteosat observations and WRF model simulation. Different colors represent different simulations (N=100, blue, N=500, green, N=1000, orange). (a) cloud liquid water path (LWP), (b) cloud droplet number concentration ($N_d$).

The overestimations of $N_d$ are due to the overestimated prescribed aerosol concentration in model setting combined with the lack of precipitating scavenging effect on aerosols in these simulations. Based on field campaign measurement, the mean total aerosol concentration ($N_a$) and $N_d$ in the ENA region in summer are ~400 $cm^{-3}$ and 80 $cm^{-3}$ respectively (Zhang et al., 2021; Wang et al., 2021; Wang et al., 2022; Zheng et al., 2024). Since the purpose of this study is to quantify aerosol-cloud interactions and cloud susceptibility in model simulations, the prescribed aerosol concentrations are designed as N=100, N=500, and N=1000 to sufficient variation in $N_a$. The simulated Nd in these simulations are lower and higher than Meteosat retrieved $N_d$ (Figure R1b). We have added the model-satellite comparison of LWP to the supplement, and added the following text to the manuscript:

"The overall overestimation of $N_d$ likely arises from the prescribed aerosol concentration used in the model configuration, combined with the absence of precipitation scavenging. For reference, the mean aerosol concentration over the ENA region during summer is approximately 400 $cm^{-3}$ (e.g., Zhang et al., 2021; Wang et al., 2021; Wang et al., 2022; Zheng et al., 2024). The model's overestimation of LWP may stem from its excessively positive LWP susceptibility in thick clouds. As shown in Figure S9, simulated LWP in the N=100 simulation agrees reasonably well with the Meteosat retrieval, with a mean value about 10% lower than observed. However, in the N=500 and N=1000 simulations, the strong positive LWP susceptibility leads to increases in LWP for clouds with LWP> 75 $gm^{-2}$, resulting in mean values 30% and 40% higher than Meteosat retrievals, respectively. (Lines 556-565)"

*- The "Data and methodology" section needs more details. For observational data, what are the specific products or variables used from satellite? What are the uncertainties of your observations and how good are they? How did you calculate Nd from Meteosat, what is the assumptions and uncertainties of the selected method on your cases?*

*For WRF model, how are the key warm cloud processes treated in your model, what are the parameterizations and what are the limitations of these treatments for your cases? What is the limitation of using a constant total aerosol number concentration throughout the domain for your model-observation comparison on LWP susceptibility? What is the default value you selected for aerosol number concentrations for your cases and are they the same for all cases? How did you quantify the Nd-LWP relationships driven by internal cloud processes and by cloud base updraft speed?*

Thanks for the questions and detailed suggestions. To address this comment, we have added the following text to the manuscript:

"The SEVIRI Meteosat cloud retrieval products are pixel-level cloud retrievals produced by NASA LaRC SatCORPS group, specifically tailored to support the ARM program over the ARM ground-based observation sites. (Lines 142-143)"

"In this study, we used the cloud mask, cloud effective radius ($r_e$), cloud optical depth ($\tau$), cloud liquid water path (LWP), cloud phase, and cloud top height variables in the product. We focus on warm boundary layer clouds with cloud top below 3km and a liquid cloud phase. The $r_e$ and $\tau$ retrievals are based on the shortwave-infrared split window technique during the daytime. Cloud LWP is derived from $r_e$ and $\tau$ using the equation: $LWP = \frac{4r_e\tau}{3Q_{ext}}$, where "$Q_{ext}$ represents the extinction efficiency and assumed constant of 2.0. Cloud mask algorithm is consistent with the CERES Ed-4 algorithm, as described in Trepte et al. (2019), where cloudy and clear pixels are distinguished based on the calculated TOA clear-sky radiance. Cloud top height is derived from the retrieved cloud effective and top temperature, together with the boundary-layer temperature profiles and lapse rate, as described in Sun-Mack et al. (2014). Cloud $N_d$ is retrieved based on the adiabatic assumptions for warm boundary layer clouds, as described in Grosvenor et al. (2018) based on the following equation:

$$N_d = \frac{\sqrt{5}}{2\pi k}\left(\frac{f_{ad}c_w\tau}{Q_{ext}\rho_w r_e^5}\right)^{1/2} \tag{1}$$

In Equation (1), $k$ represents the ratio between the volume mean radius and $r_e$, $f_{ad}$ is the adiabatic fraction, $c_w$ is the condensation rate, $Q_{ext}$ is the extinction coefficient, and $\rho_w$ is the density of liquid water (Grosvenor et al., 2018). (Lines 148-165)"

"We employed four one-way nested domains in the model, with the domain size of $27° \times 27°$, $9° \times 9°$, $3° \times 3°$, and $1° \times 1°$, and spatial resolution of 5km, 1.67 km, 0.56 km, and 190m, respectively, for d01, d02, d03, and d04 domain. The innermost domain (d04) exhibit a domain size close to most GCM grid spacing and is consistent with the spatial scale for quantification of cloud susceptibility in satellite study (e.g., Zhang et la., 2022, 2023; Qiu et al., 2024). The spatial resolution of 190 is much higher than the CPMs and close to the LES scale. All the analyses and evaluations in this study are based on output from the innermost domain (d04). (Lines 216-224) "

"To access the cloud responses to aerosol perturbations, we conduct three sets of simulations with different prescribed aerosol number concentration of N=100, 500, and 1000 $cm^{-3}$ for all 11 cases. Cloud susceptibility is quantified as the change in domain-mean cloud properties within the innermost domain at the same output time, comparing polluted and clean conditions (e.g. N=1000 vs. N=100, N=500 vs. N=100, and N=1000 vs. N=500). With constant and uniform aerosol concentration, the $N_d$-LWP relations resulting from internal cloud processes are able to be quantified within each experiment at the same output time. To minimize $N_d$-LWP relations from cloud heterogeneity and small-scale covariability and to be consistent with the quantification of cloud susceptibility in satellite observations, the pixel level model outputs are smoothed to 25-km resolution and $N_d$-LWP relations are quantified as $dln(LWP)/dln(N_d)$ using the smoothed data. (Lines 269-279)"

*- Naming of model simulations are unclear and sometimes confusing throughout the manuscript. Currently they are described with "polluted" and "clean" in comparison. This can be misleading when you switch to another set (e.g., N=500 can be "clean" compared to N=1000 but can be "polluted" compared to N=100). In addition, "clean" is also used for describing observations (Line 497) and there is also a description of "ultra-clean" (Line 560) for the N=100 simulation. I suggest a consistent name for each model configuration in the manuscript for clarity.*

Thank you for the clarification and suggestion. We have changed the naming of different simulations as N=100, N=500, and N=1000 throughout the manuscript.

*- How do different synoptic regimes affect the LWP susceptibility? You mentioned to investigate the variation of ACI across different synoptic conditions in the Introduction (Lines 117-120) and therefore chose these 11 cases, however little results and analysis are shown in this manuscript on this question.*

Thank you for the question and constructive suggestion. We compared the LWP susceptibility and the occurrence frequency of different cloud states between the "high-ridge" and "post-trough" regimes (Figures R2), only one case is available in the "weak-trough" regime (Table S1).

In our previous study using six-year ground-based observations at the ARM ENA site, Zheng et al. (2025) found that the "high-ridge" regime has significantly more single-layer stratocumulus clouds, thinner cloud depth, smaller LWP, and smaller surface rain rate compared to the "post-trough" regime. Consistently, more non-precipitating thin clouds occur in the high-ridge regime compared to the post-trough regime, with the total frequency of occurrence of 49% and 40%, respectively in the simulations (Figures R2b and d). For cloud susceptibility, the non-precipitating thin clouds in the high-ridge regime exhibit more negative LWP susceptibility compared to clouds with similar LWP and Nd in the post-trough regime, likely due to the cold dry air above clouds with the subsidence in the high-ridge regime. Additionally, the non-precipitating or slightly drizzling thick clouds in both regimes exhibit strong positive LWP susceptibilities, indicating that the model-observation discrepancy for this cloud state is consistent with different synoptic conditions and warrant further investigations in the next section.

[Figure]

Figure R2. Mean liquid water path (LWP) susceptibility from WRF simulations for (a) (b) the high-ridge regime and (c) (d) the post-trough regime. (a) (c) cloud LWP susceptibility $dln(LWP)/dln(N_d)$, (b) (d) frequency of occurrence of sample in each bin.

To address this comment, we have added Figure R2 to the supplementary material and the following text to the manuscript:

"To further examine whether these discrepancies depend on large-scale meteorological conditions, we assessed LWP susceptibility across different synoptic regimes. Because only one case is available for the "weak-trough" regime (Table S1), our comparison focuses on the "high-ridge" and the "post-trough" regimes (Figure S10). The "high-ridge" regime shows a higher occurrence of non-precipitating thin clouds than the "post-trough" regime, with total frequencies of 49% and 40%, respectively (Figures S10b, d). This more frequent non-precipitating thin cloud in the model is consistent with our previous study based on six years of ground-based observations at the ARM ENA site, which revealed that the "high-ridge" regime favors single-layer stratocumulus clouds with shallower cloud depth and smaller LWP compared to the "post-trough" regime (Zheng et al., 2025).

In addition, non-precipitating thin clouds in the "high-ridge" regime exhibit more negative LWP susceptibilities than clouds with similar LWP and $N_d$ in the "post-trough" regime. This difference in LWP susceptibility is associated with the colder and drier air above clouds under subsidence in the "high-ridge" regime, which facilitates cloud dissipation, as also demonstrated in the case study. Furthermore, non-precipitating or lightly drizzling thick clouds in both synoptic regimes manifest strong positive LWP susceptibilities, suggesting that the model-observation discrepancy for this cloud state persist regardless of synoptic conditions and therefore warrants further investigation. (Lines 566-583)"

*Many captions in this manuscript are not complete and refer to captions in another figure. I suggest to include full captions for all the figures and be clear about the data used in the figure.*

Thanks for the suggestion. The captions of the figures have been edited accordingly.

*- In Section 3.3.1 Precipitation Efficiency, there are many comparisons between model and ground-based observations for cloud with different Re and optical depth.*
*However, the current Figures 7-10 are for observations, N=100, N=500, N=1000 and each has 9 subplots categorized by Re and optical depth, making the whole section sometimes hard to follow. It might be helpful to reorganize these figures and perhaps paragraphs as well, so that observation and all model results are in the same figure for comparison. For example, Figure 7 can just contain clouds with optical depth less than 10 and the column now becomes observation, N=100, N=500, and N=1000. Or separate the figures by non-precipitating, drizzle and rain.*

Thanks for this insightful suggestion. We agree that Section 3.3.1 includes a lot of information of clouds from both observations and model outputs while clouds were further categorized by re and optical depth. This complexity sometimes makes the discussion difficult to follow. After attempting to reorganize the figures as you suggested, we found that this organization would split the discussion of model parameterization issues into three separate parts. For example, in the figure of clouds with $\tau < 10$, the issue of overestimation of precipitation at cloud top for thin clouds is reviewed and discussed. But the issue of overestimation of rain in thick clouds were not discussed until the figure of clouds with $\tau > 20$. Compared to the suggested arrangement, the current arrangement has the advantages of 1) combining the discussion of model issues and improvements together. 2) all the cloud characteristics shown in observations were combined together and discussed first, then they were compared to model simulations. As a result, we have decided to keep the current organization.

*Minor comments:*

*- Line 1: I don't think "reconciling" is accurate for the title of this manuscript. I think key processes and reasons behind the inconsistent LWP susceptibility are identified in this manuscript, but this issue is not resolved here and requires model improvement.*

Thanks for the suggestion. We agree with the reviewer that the word "reconcile" is not accurate enough. The title has been updated to "Understanding the causes of Satellite–Model Discrepancies in Aerosol–Cloud Interactions Using Near-LES Simulations of Marine Boundary Layer Clouds"

*- Lines 18-19: "largely due to" – I don't think incorrect LWP responses to aerosol perturbations is the reason but a main issue. The reasons can be poor representation of aerosol and cloud processes.*

Thanks for the suggestion. This sentence has been updated to "Aerosol–cloud interactions (ACI) remain the largest source of uncertainty in model estimates of anthropogenic radiative forcing, primarily because of deficiencies in representing aerosol–cloud microphysical processes that lead to inconsistent cloud liquid water path (LWP) responses to aerosol perturbations between observations and models."

*- Line 25: "a modest LWP decrease" to an increase in $N_d$.*
Edited.

*- Line 26: "In contrast" to? It feels coming from nowhere. If you would like to suggest that non-precipitating thin clouds have consistent LWP susceptibilities from model and observation, but not for non-precipitating thick clouds, then you need to state this clearly.*

These sentences have been edited to "Non-precipitating thin clouds exhibit a modest LWP decrease with increasing $N_d$ (mean susceptibility = −0.13), consistent in sign but weaker in magnitude than satellite estimates due to enhanced turbulent mixing and evaporation. Meanwhile, the largest model-observation discrepancy occurs in non-precipitating thick clouds, where simulated LWP susceptibilities are strongly positive while observations indicate large negative values (+0.32 vs. −0.69)."

*- Line 108: please define the abbreviation of "MBL".*
Done.

*- Lines 128-133: What are the specific cloud retrievals and what are the uncertainties of each cloud retrieval? In addition, you have the method of calculating $N_d$ from satellite mentioned at Line 386-392, but I think it will be better to move to this section. It is also useful to include version numbers of satellite product here and in the Data availability section.*

Thanks. Cloud retrieval method for each variable used in the SEVIRI Meteosat cloud retrieval product has been added. We moved the equations for calculating $N_d$ to here. Please refer to our reply to major comment #2.

*- Line 138: How was the satellite retrieval smoothed to 25-km resolution?*
The pixel-level satellite retrievals with a spatial resolution of 3km for Meteosat11 and 4km for Meteosat10 are averaged in each $25km \times 25km$ box to get the 25-km smoothed value. The 25-km cloud fraction is defined as the fraction of cloud pixels to the sum of cloudy and clear pixels in each box. As suggested by Feingold et al. (2022), $N_d$ is retrieved at pixel level and then smoothed to 25 km.

*- Line 155: "0000 UTC"*
Modified.

*- Lines 163-169: ERA5 data is not observational data but reanalysis data, therefore I don't think this should be described here under the observational data subsection. It can be put in a separate subsection, or you can change the name of this subsection to something like "Datasets" and separate into satellite data, ground-based data and reanalysis data.*
Thanks for the suggestion, the subtitle of this section has been edited to "Datasets"

*- Lines 180-182: What are the spatial resolution of the other two nested domains?*
The spatial resolution of d02 and d03 domains are 1.67 km and 0.56 km, respectively. The following text has been added to the manuscript:
"We employed four one-way nested domains in the model, with the domain size of 27° × 27°, 9° × 9°, 3° × 3°, and 1° × 1°, and spatial resolution of 5km, 1.67 km, 0.56 km, and 190m, respectively, for d01, d02, d03, and d04 domain. The innermost domain (d04) exhibit a domain size close to most GCM grid spacing and is consistent with the spatial scale for quantification of cloud susceptibility in satellite study (e.g., Zhang et la., 2022, 2023; Qiu et al., 2024)."

*- Line 186: How often is the lateral boundary condition updated?*
The lateral boundary conditions are updated every three hours.

*- Lines 189-191: How are boundary layer and clouds treated in the innermost domain?*
In the innermost domain, with the spatial resolution of 190m (close to LES resolution), the boundary layer processes and shallow cumulus clouds are resolved. We turned on the PBL scheme and shallow cumulus scheme in the d01 and d02 domains, where the Mellor–Yamada–Janjic (MYJ; Mellor and Yamada, 1982) PBL scheme and the shallow cumulus schemes (Hong and Jiang, 2018) are utilized. In d03 and d04 domains, these processes are resolved.

*- Figure 1: How does Meteosat retrieve cloud coverage and is the modelled cloud cover comparable to the Meteosat-retrieved cloud coverage? How is cloud top height defined in model output and how does Meteosat retrieve cloud top height? I suggest adding time series of Nd here. In addition, how does N=500 simulation look like?*
The Meteosat cloud coverage is defined as the fraction of cloudy pixels to the summation of cloudy and clear pixels. The cloud mask algorithm used in Meteosat cloud retrieval product is consistent with the CERES Ed4 cloud mask algorithm described in Trepte et al. (2019), where cloudy and clear pixels are distinguished based on the calculated TOA clear-sky radiance for different surface conditions, time, viewing and illumination conditions. The cloud coverage in WRF simulation is similarly a spatial fraction of cloud in the domain, where cloudy and clear of each pixel is estimated based on relative humidity and cloud water mixing ratio. As a result, the comparison of domain cloud coverage between Meteosat and WRF model is a consistent evaluation of model performance.

The cloud top height retrieval in Meteosat product is based on the cloud top temperature along with the temperature profiles and lapse rate, with an error range of 0.04 and 0.1 km over ice-free water during daytime and night time, respectively (e.g., Sun-Mack et al., 2014; Minnis et al., 2021). Cloud top height in model output is based on the "CTOPHT" variable in the model, which is estimated as the highest model level where the cloud water mixing ratio exceed a threshold. The time series of Nd has been added to Figures 1 and 3 and the corresponding discussion has been added to the text. To address this comment, the Meteosat cloud mask and cloud top height retrieval algorithms were added to the data and methodology section. Please refer to the previous major comment #2 for the added text.

*- Lines 291-292: I don't think the cloud coverage from N=100 simulation closely matches the observed cloud coverage, but underestimates the cloud cover. It will be helpful to add some numbers here as well, rather than just quantitative descriptions.*
Thanks. This sentence has been edited as "In the N=100 simulation, WRF model reproduces the overcast and precipitating stratocumulus clouds, with a domain mean cloud cover varies

between 0.90 to 0.94 from 00-13 UTC, which is slightly below that from Meteosat of 0.97 to 1.0 (Figure 1a, blue and black lines)".

*- Figure 3: Please use a full caption here rather than referring to another figure's caption. Similar to the comments for Figure 1, I suggest adding time series of $N_d$ here as well.*
Done.

*- Lines 292-294: Can you suggest the reasons behind the model failed to simulate the dissipation of clouds? And how may this bias affect the modelled LWP susceptibility?*
Thanks for your question. The lack of cloud dissipation and diurnal variation in marine boundary layer (MBL) clouds in the WRF model is likely associated with the biases in the thermodynamic profiles inherited from ERA5. As seen in Figures R1 left figure, on 21 July 2016, ARM sounding observations indicate a sharp decrease in moisture above the PBL between 14 and 20 UTC, leading to the dissipation of clouds after 14 UTC (Figure 1a). In contrast, both ERA5 and WRF simulation show a gradual decrease in specific humidity and relative humidity above the PBL from 0 to 20 UTC, resulting in a much moister layer above clouds in the model (Figures R3 middle and right). Consequently, clouds did not dissipate in the afternoon in the simulation.

On 25 July 2016, the ARM sounding observations similarly exhibit a pronounced decrease in specific humidity and relative humidity above the PBL between 14 and 24 UTC (Figure R4). In this case, the WRF simulation accurately capture the observed feature, reproducing a sharp decrease in moisture above the PBL from 14 to 24 UTC. As a result, clouds in the N=100 and N=1000 simulations dissipate from 14 to 24 UTC, consistent with satellite observation (Figure R5). This pattern also holds for other cases in the high-ridge regime, such as 22 and 28 July 2016, where the accuracy of the simulated PBL moisture variation determined whether the model captured the observed diurnal evolution of clouds (figures not shown). These cases demonstrate that the diurnal cycle of cloudiness is highly sensitive to the representation of diurnal variation in moisture as well as the moisture gradients near the inversion.

The fixed, vertically uniform aerosol concentration further contributes to the persistence of clouds by maintaining unrealistically high CCN concentrations throughout the day and suppressing precipitation. The lack of precipitation scavenging also reduces evaporative cooling and weakens cloud–PBL decoupling, inhibiting afternoon cloud breakup.
We added these discussions to the second last paragraph in Section 3.1:

"The absence of afternoon cloud dissipation in WRF simulations are likely associated with model biases in the thermodynamic structure inherited from ERA5. For example, on 21 July 2016, ARM sounding observations show a pronounced decrease in specific humidity and relative humidity above the PBL between 14 and 20 UTC (figures not shown). This sharp drying leads to cloud erosion in the observations. However, WRF simulations or ERA5 reanalysis produces only a gradual reduction in moisture from 00 to 20 UTC (Figure 2a), maintaining a moist layer above cloud top and prevent cloud breakup. On 22 July 2016, the model reproduces the moisture gradient above PBL with a warm and dry layer above, the lifted cloud top in the N=1000 simulation entrain dry air into cloud system and dissipate clouds in the afternoon (Figure 3a). On days when ERA5 accurately capture the observed moisture decrease above PBL (e.g., 25 and 28 July 2016), the model reproduces both the dissipation and evening redevelopment of clouds seen in Meteosat data (figures not shown). This indicates that the diurnal evolution of MBL clouds is highly sensitive to the representation of diurnal variation in moisture as well as the moisture gradients near the inversion.

The prescribed, vertically uniform aerosol concentration further reinforces cloud persistence by maintaining elevated CCN levels and suppressing drizzle formation. The lack of precipitation scavenging prevents cloud-base evaporative cooling and inhibits decoupling, both of which would otherwise promote afternoon cloud breakup. The implications of thermodynamic and aerosol-related biases for the estimated ACI are discussed in detail in Section 3.3.2. (Lines 436-455)"

[Figure]

Figure R3 Time series of thermodynamic profiles on 21 July 2016, for (a) potential temperature (unit: K) (b) specific humidity (unit g/kg), (c) relative humidity in (left) ARM interpolated sounding, (middle) ERA5 reanalysis, and (right) in WRF N=100 simulation.

[Figure]

Figure R4 Time series of thermodynamic profiles on 25 July 2016, for (a) potential temperature (unit: K) (b) specific humidity (unit g/kg), (c) relative humidity in (left) ARM interpolated sounding, (middle) ERA5 reanalysis, and (right) in WRF N=100 simulation.

[Figure]

Figure R5. Time series of domain-averaged cloud properties from observations and model simulation on 25 July 2016. (a) Cloud coverage, (b) cloud top height, (c) cloud liquid water path, and (d) rain-water path for N=100 (blue lines) and N=1000 (orange lines) experiments.

*- Figure 4: Please use a full caption here.*
Done

*- Figure 5: "WRF simulations" are these from all polluted versus clean simulations or just one of the sets? Are Re on these plots from the model or from satellite? Please make sure the axes are same for the model and observation plots. Currently they are different and make it difficult to compare with.*
Figure 5 has been updated with same axis ranges for all subplots. Cloud susceptibility from WRF simulations are estimated based on all three aerosol concentrations between clean and polluted experiments (e.g. N=1000 vs. N=100, N=500 vs. N=100, and N=1000 vs. N=500) to estimate the mean LWP response. In Figure 5, the $r_e = 15 \ \mu m$, $r_e = 10 \ \mu m$ isolines are estimated based on the relationships between $r_e$ and LWP and $r_e$ and $N_d$ in the satellite retrievals: $LWP = \frac{4 r_e \tau}{3 Q_{ext}}$ and $N_d = \frac{\sqrt{5}}{2 \pi k} (\frac{f_{ad} c_w \tau}{Q_{ext} \rho_w r_e^5})^{1/2}$. In Figure 6c, we show the WRF simulated mean re in the LWP-$N_d$ space.

  To address this comment, we added the following explanations on the LWP-$N_d$ space and $r_e$ isolines in the manuscript:
"Based on the relationships between $r_e$, LWP, and $N_d$ in the satellite retrievals (e.g., $LWP = \frac{4 r_e \tau}{3 Q_{ext}}$, $N_d = \frac{\sqrt{5}}{2 \pi k} (\frac{f_{ad} c_w \tau}{Q_{ext} \rho_w r_e^5})^{1/2}$), $r_e = 15$ isolines is marked in the LWP-$N_d$ parameter space as an commonly used indicator of precipitation likelihood in the satellite retrieval (e.g., Gryspeerdt et al., 2019; Toll et al., 2019; Zhang et al., 2022; Qiu et al., 2024). Based on the distinct LWP, cloud albedo and CF susceptibilities, MBL clouds are classified into three states: the precipitating clouds ($r_e > 15 \ \mu m$), the non-precipitating thick clouds ($r_e < 15 \ \mu m$, LWP> 75 $g m^{-2}$), and the non-precipitating thin clouds ($r_e < 15 \ \mu m$, LWP< 75 $g m^{-2}$) (Qiu et al., 2024). To be consistent with satellite observations, clouds in WRF simulations are classified using the same definition. (Lines 479-488)"

*- Line 382: How does the Meteosat LWP susceptibility calculated?*
We added the following text to the data and methodology section to clarify the calculation of LWP and CF susceptibilities in Meteosat data: "In the context of ACI: cloud susceptibility quantifies how sensitive a cloud property responds to change in aerosol concentration or $N_d$. To constrain the spatial-temporal variation in meteorological conditions and cloud properties, cloud susceptibility is estimated as the regression slope between $N_d$ and cloud properties within the 1° × 1° domain at each time step of satellite observations. Because of the non-linear relations between LWP and $N_d$, the LWP susceptibility is quantified in logarithm scale as $dln(LWP)/dln(N_d)$ (e.g., Gryspeerdt et al. 2019; Qiu et al., 2024), whereas cloud fraction (CF) susceptibility is quantified as $dCF/dln(N_d)$ (e.g., Kaufman et al. 2005; Chen et al., 2022; Qiu et al., 2024)."

  We also added the calculation of LWP susceptibility in Meteosat observation in Figure 5: "To evaluate model simulation, LWP susceptibility from satellite retrievals is estimated within the same domain as the model configuration for the same 11 cases (Figures 5 c, d). More specifically, LWP susceptibility is estimated as the regression slope between LWP and $N_d$ within the 1° × 1° domain at each time step of satellite observations. For precipitating clouds, LWP slightly decreases with aerosol perturbations in satellite data (Figure 5c). "

*- Line 386: What does it mean by "to be consistent with satellite observations"?*
This paragraph has been edited, please see the previous comment.

*- Line 395: I think it will be useful to add a sentence here on how you define different types of clouds: precipitating versus non-precipitating, thick versus thin.*

Thanks for the suggestions. The definitions of different cloud states have been added to the manuscript, please see the previous comment.

*- Lines 407-410: Your satellite observations for precipitating clouds are different from your simulations and previous study with long-term data. Can you suggest why? Is this because of the limitations of satellite data? Does this affect your model-satellite comparison for other clouds?*

Thanks for the question. This sentence in the paper was not accurate enough. Our satellite LWP susceptibility based on the selected 11 cases agrees well with the LWP susceptibility base on four years of satellite observations in our previous study (Qiu et al., 2024; Figure 2). As seen in Figure R6a, the LWP susceptibility is positive for precipitating thick clouds in ultra clean conditions with $N_d < 30 \ cm^{-3}$ and LWP $> 125 \ gm^{-2}$. For most of the precipitating clouds, their LWP susceptibility is negative, which is consistent with the LWP susceptibility in this study for clouds with similar properties. The slight decrease in LWP for precipitating clouds is likely due to the depletion of LWP from the sedimentation–evaporation–entrainment feedback.

We have edited this sentence in the paper as follow: "For precipitating clouds, LWP slightly decreases with aerosol perturbations in satellite data, which is consistent with the LWP susceptibility derived from four years of data in the ENA region in our previous study (Qiu et al., 2024). This decrease of LWP with increasing $N_d$ is likely associated with the depletion of LWP through sedimentation–evaporation–entrainment feedbacks, which outweigh the increase of LWP from precipitation suppression. In contrast, in model simulations, the lack of realistic evaporation-entrainment feedback results in LWP increasing primarily through precipitation suppression. The simulated LWP susceptibilities are significantly different with satellite observations at 95% confidence level for most precipitating clouds (Figure 5a). (Lines 520-528)"

[Figure]

Figure R6. Mean cloud susceptibilities for different $N_d$ and LWP bins during the daytime. (a) cloud LWP susceptibility ($dln(LWP)/dln(N_d)$), (b) cloud albedo susceptibility ($d\alpha_c/dln(N_d)$), (c) cloud fraction susceptibility ($dCF/dln(N_d)$), (d) cloud shortwave susceptibility ($-dSW_{TOA}^{up}/dln(N_d)$) weighted by the frequency of occurrence of samples of each bin, and (e) frequency of occurrence of samples in each bin. The dashed lines in (a)-(e)

indicate $r_e = 15 \mu m$ and LWP= 75 $gm^{-2}$, as thresholds for precipitation (precipitating clouds located to the left of the line) and thick clouds (with LWP > 75 $gm^{-2}$). The defined three clouds states are noted in (a). (Figure was adapted from Qiu et al., 2024)

*- Line 429-430: I don't think Figure 5 show that the model results agree with Meteosat observations for an increase in LWP in precipitating clouds (Meteosat suggest a decrease).*

Yes, I agree with you that this sentence was not accurate, and it has been edited.

*- Line 434-435: If the modelled LWP response is showing large discrepancy compared to observations, this is not indicating the robustness of the results. Please explain in detail on the reasons why you suggest that the model results are robust.*

Thanks for the clarification. This sentence has been modified. The agreements with previous model results indicate consistency instead of robustness of our model results.

*- Figure 6: It is confusing here that the $r_e$ dashed lines across different $r_e$ contour colors in (a) and (c). Please be clear about how each effective radius is calculated or derived in (a), (c) and the dash line.*

Thank you for pointing out this unclear point. In satellite observations, both LWP and $N_d$ are retrieved as a function of function of $r_e$ and $\tau$ . In Figure 6a, the $r_e$ dashed lines are based on the relationships between $r_e$, LWP, and $N_d$ in the satellite retrievals (e.g., $LWP = \frac{4 r_e \tau}{3 Q_{ext}}$,

$N_d = \frac{\sqrt{5}}{2\pi k}(\frac{f_{ad} c_w \tau}{Q_{ext}\rho_w r_e^5})^{1/2}$ ), and are used as indications of precipitation in the figure. As a result, the $r_e$ dash lines agree with the $r_e$ contour in Figure 6a. To use the same classification of precipitation in the model as the satellite observations, in Figure 6c, the $r_e$ dashed lines are based on the same relationships between $r_e$, LWP, and $N_d$ in the satellite retrievals, while the $r_e$ contour is from model output. As model underestimate $r_e$ compared with satellite observations, the dash lines cross different contour colors as in Figure 6a.

*- Lines 493-495: Frequencies from satellite data only sum to 90.6%, what and where are the rest 9.4%? In addition, can you explain more on why the selected cases are representative just based on the frequencies?*

Thanks for pointing out the mistake in the frequency of satellite data. We have corrected the mistake in the calculation of frequency, the occurrence frequency for different cloud states have been updated to 22.2%, 55.6%, and 22.2%, respectively. The occurrence frequency of precipitating, non-precipitating thin and non-precipitating thick clouds based on the selected cases align well with the occurrence frequency of cloud states in ARM data based on six-year of observations, with non-precipitating thin clouds the dominate cloud type and thick clouds the least frequent cloud type, suggesting that the selected cases are representative of the typical distribution of MBL cloud types in the ENA region in summer.

*- Line 497 and others: what does "clean condition" mean here? You use "clean" to describe both simulations and $R_e$ condition in your figures in this section, which is confusing during reading.*

Thanks for the suggestion and clarification. The clean condition here indicates under clean environment in the observations when cloud $r_e$ is greater than 15 μm. The naming of model simulations has been edited to N=100, N=500, and N=1000 in the manuscript instead of using clean and polluted.

*- Lines 506-507: "likely due to mixing and evaporation" – can you be more specific on this?*

As seen in Figure 7h, the maximum frequency of radar reflectivity shows a decrease of signal towards cloud base, which is likely due to the mixing and entrainment from cloud base which lead to the evaporation of cloud and rain drops.

- *Figures 8-10: Please use full captions for these figures.*
- Done
-

*Lines 558-559: I can see that DSD is compared by using percentages of R$_e$ categorizes,*
*but it may be helpful and clearer to compare full DSD from different model simulations and*
*observations for clouds with different optical depths.*

Thanks for the suggestion. As we don't have the observed DSD data from filed campaign for
most of our cases, I compare the DSD in one of our case with the measured DSD in Yeom et al.
(2022) study based on the ACE-ENA campaign data on a different day.

[Figure]

Figure R7: Cloud drop size distribution from WRF simulation on 21 July 2016.

[Figure]

Figure R8 Average drop size distribution of bin number concentration of undiluted (solid lines) and diluted (dash lines) section at each horizontal penetration for (a) P1 (b) P2 flight on 18 July 2017 during the ACE-ENA field campaign. (Figure adopted from Yeom et al., 2022)

As shown in Figure R7, the DSD from N=1000 simulation is wider than that from N=100 simulation. This result is consistent with the parametrization of DSD in the Morrison scheme, where the dispersion parameter is proportional to $N_d$. As seen from Figures R7 and R8, the simulated DSD is wider than observation, especially for the N=1000 simulation.

The DSD of the model parameterization is one of our hypotheses to explain the excessive rain production in clouds with $r_e > 15\ \mu m$. Our results between different simulations show consistent characteristics with the DSD parameterization in the Morrison scheme, which support this hypothesis. For example, the overestimation of rain is more and more sever in the N=500 and N=1000 simulations comparing with the N=100 simulation for clouds with $r_e > 15\ \mu m$. However, as most of our cases don't have direct field campaign data, it is difficult to validate the DSD in the model.

*- Lines 591-599: The description of DSD in the model is better to be put in the Data and Methodology section along with the descriptions of other treatments of warm cloud processes.*
Thanks, this part has been moved to the method section.

*- Lines 636-638: The cloud tops are defined differently in ARM observations and in the model. Since you have the model radar simulator, why not using the same definition here based on the radar reflectivity profile for observed and modelled cloud tops?*
Following your suggestion, we calculated the domain mean cloud top height in WRF simulation based on the radar reflectivity profiles from the radar simulator and use the same reflectivity threshold to retrieve cloud top height as in ARM observation (reflectivity > -40 dBZ). Figure R9 compares the PDF between the cloud top height retrieved using reflectivity > -40 dBZ with that retrieved using cloud water mixing ratio > 0.001 g/kg for all 11 cases and all three aerosol concentrations. As seen in Figure R9, the retrieved cloud-top height is consistent between the two methods, with a difference in mean value of less than 40m. Therefore, the large discrepancy in cloud top RH between WRF simulations and ARM observations are not due to the different cloud top retrieval methods.

We have added the related discussion to the paper: "We further compare the cloud-top heights in WRF simulations defined using cloud water mixing ratio and radar reflectivity profiles with $Z_e > -40$ dBZ. The two approaches yield nearly identical results, with a mean difference of less than 40m (figure not shown). (Lines 855-858)"

[Figure]

Figure R9. PDF of WRF simulated cloud top height retrieved using cloud water mixing ratio threshold (blue line) and using cloud radar reflectivity threshold (red line).

*- Lines 644-648: I recommend the authors on considering the spatial representation issue and it will be helpful to describe how the temporal representation issue is treated, e.g., what are the model output time for comparing cloud top RH with the sounding observations?*
Thanks for the question. The spatial and temporal representation issues is resolved by calculating the model output over a 10km × 10km grid box centered at the ARM ENA site for each sounding time. Given the ~1.2-1.4 km mean cloud-top height for MBL clouds at ENA during summer, and the balloon sounding rise at ~1m/s speed, it takes ~1200s-1400s to reach the cloud top. With the prevailing wind speed of 7 m/s, the balloon travels ~8 to 10km horizontally. Therefore, we averaged the WRF pixel-level output over a 10km × 10km grid box centered at the ARM ENA site for each sounding time (usually at 12:00 UTC and 0:00 UTC each day).

*- Figure 12: What is the shaded area for?*
The solid blue line is the median value for each RH bin and the shaded area shows the lower and upper 25th percentiles of cloud susceptibility in each cloud top RH bin. This information has been added to figure captions of Figures 12 and 13.

*- Line 656: "in the simulations"- are these for all simulations with all aerosol number concentration or specific ones? Does the dependence of these cloud susceptibilities on cloud top relative humidity change when using different sets of simulations (e.g., between N=1000 vs. N=100 and N=1000 vs. N=500)?*
Thanks for the clarification question. Yes, Figure 12 shows the mean relationships between cloud top RH and cloud susceptibilities calculated based on all simulations (e.g. N=1000 vs. N=100, N=500 vs. N=100, and N=1000 vs. N=500). The cloud-top RH is also based on all simulations using cloud-top heights in N=100, N=500, and N=1000 simulations. Figures R10 and 11 show these relationships between N=100 and N=500, N=100 vs. N=1000, respectively. As seen in these figures, the positive relationships between cloud-top RH and cloud susceptibility are consistent across different aerosol concentrations.
To address this comment, we have added the following text to the manuscript:

"Figure 12 shows the mean relationship between clout-top RH and cloud susceptibilities calculated based on domain mean values for all three simulations (e.g. N=1000 vs. N=100, N=500 vs. N=100, and N=1000 vs. N=500). The cloud top RH is the domain mean RH value at ~100m above cloud top for all simulations. As seen in Figure 12a, we find a positive correlation between cloud-top RH and LWP susceptibility in the simulations, which is consistent with cloud responses shown in case study where a dry layer above cloud promotes evaporation and decrease LWP. Additionally, these positive relationships are consistent among different aerosol concentrations (e.g., N=1000 vs. N=100 or N=500 vs. N=100; figures not shown). (Lines 870-878)"

[Figure]

Figure R10. Dependence of (a) LWP susceptibility and (b) CF susceptibility on cloud top relative humidity in WRF simulations between N=100 vs. N=500. The solid blue line shows the median value of each RH bins and the shaded area shows the lower and upper 25th percentiles.

[Figure]

Figure R11. Dependence of (a) LWP susceptibility and (b) CF susceptibility on cloud top relative humidity in WRF simulations between N=100 vs. N=1000. The solid blue line shows the median value of each RH bins and the shaded area shows the lower and upper 25th percentiles.

*- Figure 13: It will be helpful to have vertical lines where buoyancy flux difference equals to 0 as well on the plot. Similar to the comment on Figure 12, what is the shaded area for?*
Figure 13 and the figure caption have been updated.

*Figure 14: Are these from simulations with all different aerosol concentrations?*
Yes, results shown in Figure 14 are based on the 11 cases and all three aerosol concentrations.
*- Figure 15: Please use a full caption.*
- Done.
*- Lines 745-754: I think several references are missing here in this first paragraph when mentioning the findings from previous studies.*
Thanks. The references have been added to the manuscript.
*- Lines 762-763: I suggest adding the LWP bias here rather than using "generally match".*
Thanks. This sentence has been updated to "The simulated MBL clouds generally match the satellite observation in domain mean cloud coverage and mesoscale

organization (Figures 1, 3, S2-S4), while the model may struggle to capture the diurnal evolution of clouds, especially the dissipation of clouds in the afternoon. Model overestimate cloud LWP, especially in the polluted runs and underestimated cloud top height compared to satellite retrievals."

- Lines 778: *Are there any other potential reasons for the LWP bias and what's the reason that you suggest the lack of precipitation scavenging feedback on aerosols is likely the cause here?*

The overestimations of $N_d$ are due to the overestimated prescribed aerosol concentration in model setting combined with the lack of precipitating scavenging effect in WRF model. The overestimation of LWP is likely due to the positive LWP susceptibility for thick clouds, as shown in Figure R1a. This sentence has been updated to "Meanwhile, the non-precipitating thick clouds are the dominant cloud state in the model, with a total frequency of 49%, compared to a 15.7% frequency of occurrence in satellite observations. The overestimation of $N_d$ arise from the overestimated aerosol concentration in the configuration, combined with the absence of precipitation scavenging in the model. The overestimation of LWP is due to the positive LWP susceptibility in thick clouds where LWP in N=100 simulation show good agreement with satellite retrievals (Figure S9). (Lines 1034-1038)"